# SlowFast-VGen: Slow-Fast Learning for Action-Driven Long Video Generation

**Yining Hong[1], Beide Liu[1]†, Maxine Wu[1]†, Yuanhao Zhai[3], Kai-Wei Chang[1],**
**Linjie Li[2], Kevin Lin[2], Chung-Ching Lin[2], Jianfeng Wang[2], Zhengyuan Yang[2,††],**
**Yingnian Wu[1,††], Lijuan Wang[2,††]**
[1] UCLA, [2] Microsoft Research, [3] State University of New York at Buffalo
† Co-second Contribution, †† Equal Advising

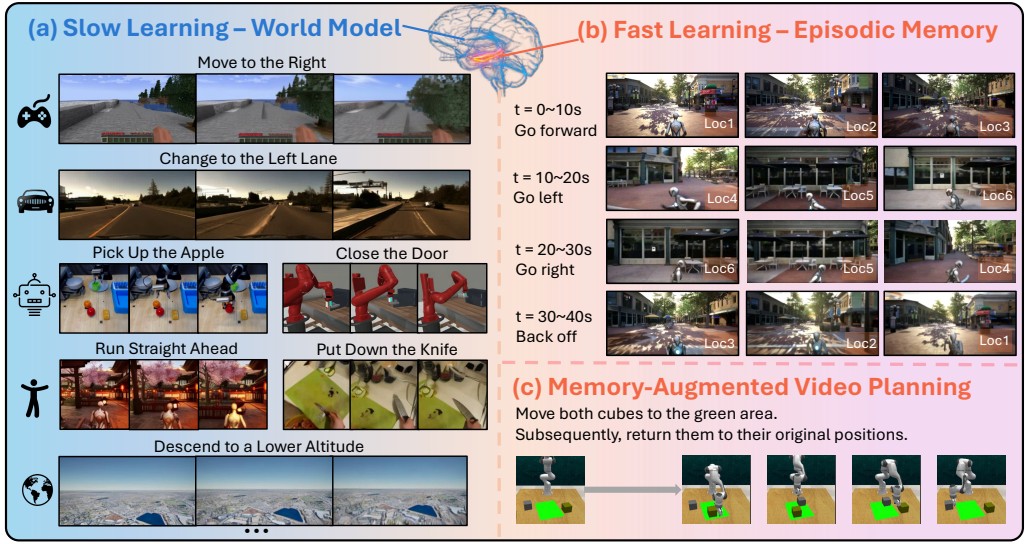

Figure 1: We propose SLOWFAST-VGEN, a dual-speed action-driven video generation system that mimics the complementary learning system in human brains. The slow learning phase (a) learns an approximate world model that simulates general dynamics across a diverse set of scenarios. The fast learning phase (b) stores episodic memory for consistent long video generation, *e.g.,* generating the same scene for "Loc1" after traveling across different locations. Slow-fast learning also facilitates long-horizon planning tasks (c) that require the efficient storage of long-term episodic memories.

## Abstract

Human beings are endowed with a complementary learning system, which bridges the slow learning of general world dynamics with fast storage of episodic memory from a new experience. Previous video generation models, however, primarily focus on slow learning by pre-training on vast amounts of data, overlooking the fast learning phase crucial for episodic memory storage. This oversight leads to inconsistencies across temporally distant frames when generating longer videos, as these frames fall beyond the model's context window. To this end, we introduce SLOWFAST-VGEN, a novel dual-speed learning system for action-driven long video generation. Our approach incorporates a masked conditional video diffusion model for the slow learning of world dynamics, alongside an inference-time fast learning strategy based on a temporal LoRA module. Specifically, the fast learning process updates its temporal LoRA parameters based on local inputs and outputs, thereby efficiently storing episodic memory in its parameters. We further propose a slow-fast learning loop algorithm that seamlessly integrates the inner fast learning loop into the outer slow learning loop, enabling the recall of prior multi-episode experiences for context-aware skill learning. To facilitate the slow learning of an approximate world model, we collect a large-scale dataset of 200k videos with language action annotations, covering a wide range of scenarios. Extensive experiments show that SLOWFAST-VGEN outperforms baselines across various metrics for action-driven video generation, achieving an FVD score of 514

compared to 782, and maintaining consistency in longer videos, with an average of 0.37 scene cuts versus 0.89. The slow-fast learning loop algorithm significantly enhances performances on long-horizon planning tasks as well. Project Website: `https://slowfast-vgen.github.io`

# 1 INTRODUCTION

Human beings are endowed with a complementary learning system (McClelland et al., 1995). The slow learning paradigm, facilitated by the neocortex, carves out a world model from encyclopedic scenarios we have encountered, enabling us to make decisions by anticipating potential outcomes of actions (Schwesinger, 1955), as shown in Figure 1 (a). Complementing this, the hippocampus supports fast learning, allowing for rapid adaptation to new environments and efficient storage of episodic memory (Tulving, 1983), ensuring that long-horizon simulations remain consistent (*e.g.,* in Figure 1 (b), when we travel across locations 1 to 6 and return, the scene remains unchanged.)

Recently, riding the wave of success in video generation (Blattmann et al., 2023a), researchers have introduced action-driven video generation that could generate future frames conditioned on actions (Yang et al., 2024; Wang et al., 2024a; Hu et al., 2023; Wu et al., 2024), mimicking the slow learning paradigm for building the world model. However, these models often generate videos of constrained lengths (*e.g.,* 16 frames for 4 secs (Xiang et al., 2024; Wu et al., 2024)), limiting their ability to foresee far into the future. Several works focus on long video generation, using the outputs of one step as inputs to the next (Harvey et al., 2022; Villegas et al., 2022; Chen et al., 2023; Guo et al., 2023a; Henschel et al., 2024). Due to the lack of fast learning, these models often do not memorize trajectories prior to the current context window, leading to inconsistencies in long-term videos.

In this paper, we propose SLOWFAST-VGEN, which seamlessly integrates slow and fast learning in a dual-speed learning system for action-driven long video generation. For the slow learning process, we introduce a masked conditional video diffusion model that conditions on language inputs denoting actions and the preceding video chunk, to generate the subsequent chunk. In the inference time, long video generation can be achieved in a stream-in fashion, where we consecutively input our actions for the current context window, and the succeeding chunk can be generated. To enable the model to memorize beyond the previous chunk, we further propose a fast learning strategy, which rapidly adapts to new contexts and stores episodic memory, enabling the model to maintain coherence over extended sequences. By combining slow learning and fast learning, SLOWFAST-VGEN can generate consistent, action-responsive long videos, as shown in Figure 1 (a) and (b).

A key challenge for fast learning in long video generation resides in maintaining consistency across long sequences, while avoiding computationally intensive or memory-inefficient strategies for storing lengthy video sequences. To address this, we propose a novel inference-time fast learning strategy that stores episodic memory in Low-Rank Adaptation (LoRA) parameters. Specifically, we draw inspiration from Temporary LoRA (Wang et al., 2024b), originally developed for long text generation, and extend this concept for fast learning in video generation. In its original form, TEMP-LORA embeds contextual information in a temporary LoRA module for language modeling by progressively generating new text chunks based on inputs and training on these input-output pairs. Our approach modifies the update mechanism of TEMP-LORA parameters by discarding the input-to-output format. At each context window, our TEMP-LORA for fast learning operates as follows: we generate a new video chunk conditioned on both the input chunk and the corresponding action, then concatenate the input and output into a seamless temporal continuum. This sequence undergoes a noise injection process, creating a temporally coherent learning signal. We then train the denoising UNet on this noise-augmented, action-agnostic sequence. This method emphasizes memorizing entire trajectories rather than focusing on immediate input-output streams or transitions. By leveraging the forward diffusion and reverse denoising processes over the concatenated sequence, we effectively consolidate sequential episodic memory in the TEMP-LORA parameters of the UNet, enabling it to maintain coherence across extended video sequences. Qualitative examples in Figure 4 and supplementary materials demonstrate that fast learning enhances the consistency, smoothness, and quality of long videos, as well as enabling the fast adaptation to test scenes.

While our fast learning module excels at rapidly adapting to new contexts and storing episodic memory, there exist specific context-aware tasks that require learning from all previous long-term episodes rather than just memorizing individual ones. For instance, in the planning section of Figure 1 (c), the task requires not only recalling the long-term trajectory within each episode, but also

leveraging prior experiences across multiple episodes to develop the general skill of returning the cubes. Therefore, we introduce the slow-fast learning loop by integrating TEMP-LORA into the slow learning process. The inner fast learning loop uses TEMP-LORA to swiftly adapt to each episode, preparing a dataset of inputs, outputs, and corresponding TEMP-LORA parameters for the slow learning process. The outer slow learning loop then leverages the multi-episode data to update the model's core weights with frozen TEMP-LORA parameters.

To further boost slow learning and enable action-driven generation with massive data about world knowledge, we collect a large-scale video dataset comprising 200k videos paired with language action annotations. This dataset covers a myriad of tasks, including Games, Unreal Simulations, Human Activities, Driving, and Robotics. Experiments show that our SLOWFAST-VGEN outperforms baseline models on a series of metrics with regard to both slow learning and fast learning, including FVD, PSNR, SSIM, LPIPS, Scene Cuts, and our newly-proposed Scene Return Consistency (SRC). Furthermore, results on two carefully-designed long-horizon video planning tasks demonstrate that the slow-fast learning loop enables both efficient episodic memory storage and skill learning.

To sum up, our contributions are listed as below:
- We introduce SLOWFAST-VGEN, a dual-speed learning system for action-driven long video generation, which incorporates episodic memory from fast learning into an approximate world model, enabling consistent and coherent action-responsive long video generation.
- We develop an inference-time fast learning strategy that enables the storage of long-term episodic memory within LoRA parameters for long video generation. We also introduce a slow-fast learning loop that integrates the fast learning module into the slow learning process for context-aware skill learning on multi-episode data.
- We develop a masked conditional video diffusion model for slow learning and create a large-scale dataset with 200k videos paired with language action annotations, covering diverse categories.
- Experiments show that SLOWFAST-VGEN outperforms various baselines, achieving an FVD score of 514 versus 782, and averaging 0.37 scene cuts compared with 0.89. Qualitative results highlight the enhanced consistency, smoothness and quality in generated long videos. Additionally, results on long-horizon planning tasks prove that the slow-fast learning loop can effectively store episodic memory for skill learning.

## 2 RELATED WORKS

**Text-to-Video Generation** Nowadays, research on text-conditioned video generation has been evolving fast (Ho et al., 2022b;a; Singer et al., 2022; Luo et al., 2023; Esser et al., 2023; Blattmann et al., 2023b; Zhou et al., 2023; Khachatryan et al., 2023; Wang et al., 2023), enabling the generation of short videos based on text inputs. Recently, several studies have introduced world models capable of generating videos conditioned on free-text actions (Wang et al., 2024a; Xiang et al., 2024; Yang et al., 2024). While these papers claim that a general world model is pretrained, they often struggle to simulate dynamics beyond the context window, thus unable to generate consistent long videos. There are also several works that formulate robot planning as a text-to-video generation problem (Du et al., 2023; Ko et al., 2023). Specifically, Du et al. (2023) trains a video diffusion model to predict future frames and gets actions with inverse dynamic policy, which is followed by this paper.

**Long Video Generation** A primary challenge in long video generation lies in the limited number of frames that can be processed simultaneously due to memory constraints. Existing techniques often condition on the last chunk to generate the next chunk(Harvey et al., 2022; Villegas et al., 2022; Chen et al., 2023; Guo et al., 2023a; Henschel et al., 2024; Zeng et al., 2023; Ren et al., 2024), which hinders the model's ability to retain information beyond the most recent chunk, leading to inconsistencies when revisiting earlier scenes. Some approaches use an anchor frame (Yang et al., 2023; Henschel et al., 2024) to capture global context, but this is often insufficient for memorizing the whole trajectory. Other methods generate key frames and interpolate between them (He et al., 2023; Ge et al., 2022; Harvey et al., 2022; Yin et al., 2023), diverging from how world models simulate future states through sequential actions, limiting their suitability for real-time action streaming, such as in gaming contexts. Additionally, these methods require long training videos, which are hard to obtain due to frequent shot changes in online content. In this paper, we propose storing episodic memory of the entire generated sequence in LoRA parameters during inference-time fast learning, enabling rapid adaptation to new scenes while retaining consistency with a limited set of parameters.

**Low-Rank Adaptation** LoRA (Hu et al., 2021) addresses the computational challenges of fine-tuning large pretrained language models by using low-rank matrices to approximate weight changes,

reducing parameter training and hardware demands. It enables efficient task-switching and personalization without additional inference costs, as the trainable matrices integrate seamlessly with frozen weights. In this work, we develop a TEMP-LORA module, which adapts quickly to new scenes during inference and stores long-context memory in its parameters.

## 3 SLOWFAST-VGEN

In this section, we introduce our SLOWFAST-VGEN framework. We first present the masked diffusion model for slow learning and the dataset we collected (Sec 3.1). Slow learning enables the generation of new chunks based on previous ones and input actions, while lacking memory retention beyond the most recent chunk. To address this, we introduce a fast learning strategy, which can store episodic memory in LoRA parameters (Sec 3.2). Moving forward, we propose a slow-fast learning loop algorithm (Sec 3.3) that integrates TEMP-LORA into the slow learning process for context-aware tasks that require knowledge from multiple episodes, such as long-horizon planning. We also illustrate how to apply this framework for video planning (Sec 4.2) and conduct a thorough investigation and comparison between our model and complementary learning system in cognitive science, highlighting parallels between artificial and biological learning mechanisms (Sec 3.5).

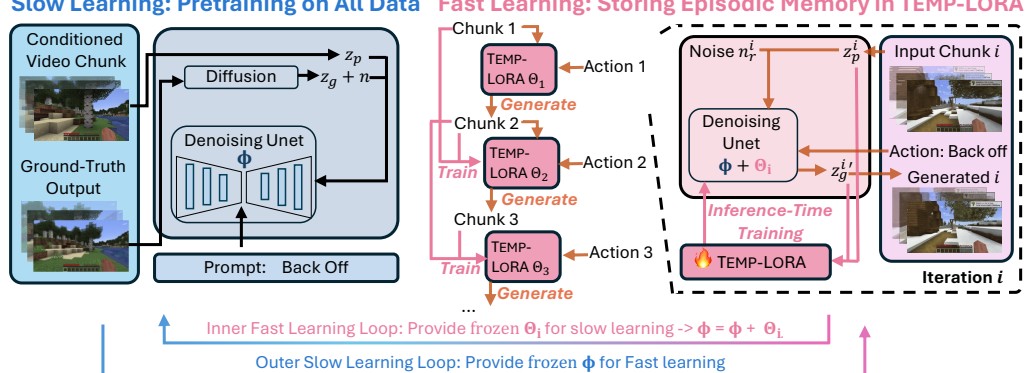

Figure 2: **SLOWFAST-VGEN Architecture**. The left side illustrates the slow learning process, pretraining on all data with a masked conditional video diffusion model. The right side depicts the fast learning process, where TEMP-LORA stores episodic memory during inference. Stream-in actions guide the generation of video chunks, with TEMP-LORA parameters updated after each chunk. In our slow-fast learning loop algorithm, the inner loop performs fast learning, supplying TEMP-LORA parameters from multiple episodes to the slow learning process, which updates slow learning parameters $\Phi$ based on frozen fast learning parameters.

### 3.1 SLOW LEARNING

#### 3.1.1 MASKED CONDITIONAL VIDEO DIFFUSION

We develop our slow learning model based on the pre-trained ModelScopeT2V model (Wang et al., 2023), which generates videos from text prompts using a latent video diffusion approach. It encodes training videos into a latent space $z$, gradually adding Gaussian noise to the latent via $z_t = \sqrt{\bar{\alpha}_t} z_0 + \sqrt{1 - \bar{\alpha}_t}\epsilon$. A Unet architecture, augmented with spatial-temporal blocks, is responsible for denoising. Text prompts are encoded by the CLIP encoder (Radford et al., 2021).

To better condition on the previous chunk, we follow Voleti et al. (2022) for masked conditional video diffusion, conditioning on past frames in the last video chunk to generate frames for the subsequent chunk, while applying masks on past frames for loss calculation. Given $f_p$ past frames and $f_g$ frames to be generated, we revise the Gaussian diffusion process to:

$$z_{t,:f_p} = z_{0,:f_p}$$
$$z_{t,f_p:(f_p+f_g)} = \sqrt{\bar{\alpha}_t} z_{0,f_p:(f_p+f_g)} + \sqrt{1 - \bar{\alpha}_t}\epsilon \qquad (1)$$
$$z_t = z_{t,:f_p} \oplus z_{t,f_p:(f_p+f_g)}$$

where $z_{t,:f_p}$ is the latent of the first $f_p$ frames at diffusion step $t$, which is clean as we do not add noise to the conditional frames. $z_{t,f_p:(f_p+f_g)}$ corresponds to the latent of the ground-truth output frames, which we add noise to for conditional generation of the diffusion process. We concatenate ($\oplus$) them together and send them all into the UNet. We then get the noise predictions out of the

UNet, and apply masks to the losses corresponding to the first $f_p$ frames. Thus, only the last $f_g$ frames are used for calculating loss. The final loss is:

$$-0.7em L(\Phi) = \mathbb{E}_{t, z_0 \sim p_{\text{data}}, \epsilon \sim \mathcal{N}(0,1), c} \left[ ||\epsilon - \epsilon_\Phi(z_t[f_p : (f_p + f_g)], t, c)||_2^2 \right] -0.7em \qquad (2)$$

Here, $c$ represents the conditioning (e.g., text), and $\Phi$ denotes the UNet parameters. Our model is able to handle videos of arbitrary lengths smaller than the context window size (*i.e.*, arbitrary $f_p$ and $f_g$). For the first frame image input, $f_p$ equals 1.

### 3.1.2 DATASET COLLECTION

Our slow learning dataset consists of 200k data, with each data point in the format of ⟨input video chunk, input free-text action, output video chunk⟩. The dataset can be categorized into 4 domains :

- **Unreal**. We utilize the Unreal Game Engine (Epic Games) for data collection, incorporating environments such as Google 3D Tiles, Unreal City Sample, and various assets purchased online. We introduce different agent types (e.g., human agents and droids) and use a Python script to automate action control. We record videos from both first-person and third-person perspectives, capturing keyboard and mouse inputs as actions and translating them into text (e.g., "go left").
- **Game**. We manually play Minecraft, recording keyboard and mouse inputs and capturing videos.
- **Human Activities**. We include the EPIC-KITCHENS (Damen et al., 2018; 2022) dataset, which comprises extensive first-person (egocentric) vision recordings of daily activities in the kitchen.
- **Robot**. We use several datasets from OpenX-Embodiment (Collaboration et al., 2024), as well as tasks from Metaworld (Yu et al., 2021) and RLBench (James et al., 2019). As most robot datasets consist of short episodes (where each episode is linked with one language instruction) rather than long videos with sequential language inputs, we set $f_p$ to 1 for these datasets.
- **Driving**. We utilize the HRI Driving Dataset (HDD) (Ramanishka et al., 2018), which includes 104 hours of real human driving. We also include driving videos generated in the Unreal Engine.

### 3.2 FAST LEARNING

The slow learning process enables the generation of new video chunks based on action descriptions. A complete episode can thus be generated by sequentially conditioning on previous outputs. However, this method does not ensure the retention of memory beyond the most recent chunk, potentially leading to inconsistencies among temporally distant segments. In this section, we introduce the novel fast learning strategy, which can store episodic memory of all generated chunks. We begin by briefly outlining the generation process of our video diffusion model. Subsequently, we introduce a temporary LoRA module, TEMP-LORA, for storing episodic memory.

**Generation** Each iteration $i$ consists of $T$ denoising steps ($t = T \dots 1$). We initialize new chunks with random noise at $z_T^i$. During each step, we combine the previous iteration's clean output $z_0^{i-1}$ with the current noisy latent $z_t^i$. This combined input is fed into a UNet to predict and remove noise, producing a less noisy version for the next step. We focus only on the newly generated frames, masking out previous-iteration outputs. This process repeats until the iteration's final step, yielding a clean latent $z_0^i$. The resulting $z_0^i$ becomes input for the next iteration. The generation process works directly with the latent representation, progressively extending the video without decoding and re-encoding from pixel space between iterations.

**TEMP-LORA** Inspired by Wang et al. (2024b), we utilize a temporary LoRA module that embeds the episodic memory in its parameters. TEMP-LORA was initially designed for long text generation, progressively generating new text chunks based on inputs, and use the generated chunk as ground-truth to train the model conditioned on the input chunk. We improve TEMP-LORA for video generation to focus on memorizing entire trajectories rather than focusing on immediate input-output streams. Specifically, after the generation process of iteration $i$, we use the concatenation of the input latent and output latent at this iteration to update the TEMP-LORA parameters.

$$z_0^{i\prime} = z_0^{i-1} \oplus z_0^i; \quad z_t^{i\prime} = \sqrt{\bar{\alpha}_t} z_0^{i\prime} + \sqrt{1 - \bar{\alpha}_t} \epsilon$$

$$L(\Theta_i | \Phi) = \mathbb{E}_{t, z_0^{i\prime}, \epsilon \sim \mathcal{N}(0,1)} \left[ ||\epsilon - \epsilon_{\Phi + \Theta_i}(z_t^{i\prime}, t)||_2^2 \right] \qquad (3)$$

Specifically, we concatenate the clean output from the last iteration $z_0^{i-1}$ (which is also the input of the current iteration) and the clean output from the current iteration $z_0^i$ to construct a temporal continuum $z_0^{i\prime}$. Then, we add noise to the whole $z_0^{i\prime}$ sequence, which results in $z_t^{i\prime}$ at noise diffusion step $t$. Note that here we do not condition on clean $z_0^{i-1}$ anymore, and exclude text conditioning to

---

**Algorithm 1** SLOWFAST-VGEN Algorithm

---

**(a) Fast Learning**

   **Input:**    First frame of video sequence $\mathcal{X}_0$, total generating iterations $\mathcal{I}$, video diffusion model with frozen pretrained weights $\Phi$ and LoRA parameters $\Theta_0$, fast learning learning rate $\alpha$

   **Output:**    Long Video Sequence $\mathcal{Y}$

1:  $\mathcal{X}_0 = \text{VAE\_ENCODE}(\mathcal{X}_0); \mathcal{Y} \leftarrow X_0$                        {Encode into latents}

2:  **for** $i$ in 0 to $\mathcal{I} - 1$ **do**

3:        **// Generate the sequence in the current context window**

4:    **if** $i \neq 0$ **then**

5:       $\mathcal{X}_i \leftarrow \mathcal{Y}_{i-1}$

6:    **end if**

7:    $\mathcal{C}_i \leftarrow \text{User\_Input}(i)$                {Action conditioning acquired through user interface input}

8:    $\mathcal{Y}_i = (\Phi + \Theta_i)(\mathcal{X}_i, C_i)$

9:    $\mathcal{Y} = \mathcal{Y} \oplus \mathcal{Y}_i$                {Concatenate the output latents to the final sequence latents}

10:    **// Use the input and output in the context window to train TEMP-LORA**

11:    $\mathcal{X}_i = \mathcal{X}_i \oplus \mathcal{Y}_i$         {Concatenate input and output to prepare TEMP-LORA training data}

12:    Sample Noise $\mathcal{N}$ on the whole $\mathcal{X}_i$ sequence

13:    Calculate Loss on the whole $\mathcal{X}_i$ sequence

14:    $\Theta_{i+1} \leftarrow \Theta_i - \alpha \cdot \nabla_\theta \text{Loss}$

15:  **end for**

16:  $\mathcal{Y} = \text{VAE\_DECODE}(\mathcal{Y})$                    {Decode into video}

17:  **return** $\mathcal{Y}$

**(b) Slow-Fast Learning Loop**

   **Definition:**    task-specific slow learning weights $\Phi$, task-specific dataset $D$, LoRA parameters of all episodes $\Theta$, slow learning learning rate $\beta$, slow-learning dataset $D_s$

   **// Slow Learning Loop**

1:  **while** not converged **do**

2:    $D_s \leftarrow \emptyset$ //prepare dataset for slow learning

3:    **for** each sample $(x, \text{episode})$ in $D$ **do**

4:      **// Fast Learning Loop**

5:      Suppose episode could be divided into $\mathcal{I}$ short sequences: $\mathcal{X}_i^e$ for i in 0 to $\mathcal{I} - 1$

6:      Initialize TEMP-LORA parameters for this episode $\Theta_0^e$

7:      **for** $i$ in 0 to $\mathcal{I} - 1$ **do**

8:        $D_s = D_s \cup \{X_i^e, X_{i+1}^e, \Theta_i^e\}$           {$X_{i+1}^e$ is the ground-truth output of input $X_i^e$}

9:        Fix $\Phi$ and update $\Theta_i^e$ using fast learning algorithm

10:      **end for**

11:    **end for**

12:    // Use the $D_s$ dataset for slow learning update

13:    **for** $\{X_i^e, X_{i+1}^e, \Theta_i^e\}$ in $D_s$ **do**

14:      $\Phi_i^e = \Phi + \Theta_i^e$

15:      Calculate Loss based on the model output of input $X_i^e$, and ground-truth output $X_{i+1}^e$

16:      Fix $\Theta_i^e$ and update $\Phi$ only: $\Phi \leftarrow \Phi - \beta \cdot \nabla_\Phi \text{Loss}$

17:    **end for**

18:  **end while**

---

focus on remembering full trajectories rather than conditions. Then, we update the TEMP-LORA parameters of the UNet $\Theta_i$ using the concatenated noise-augmented, action-agnostic $z_0^{i\prime}$. Through forward diffusion and reverse denoising, we effectively consolidate sequential episodic memory in the TEMP-LORA parameters. The fast learning algorithm is illustrated in Algorithm 1 (a).

### 3.3 SLOW-FAST LEARNING LOOP WITH TEMP-LORA

Previously, we develop TEMP-LORA for inference-time training, allowing for rapid adaptation to new contexts and the storage of per-episode memory in the LoRA parameters. However, some specific context-aware tasks require learning from all collected long-term episodes rather than just memorizing individual ones. For instance, solving a maze involves not only recalling the long-term trajectory within each maze, but also leveraging prior experiences to generalize the ability to navigate and solve different mazes effectively. Therefore, to enhance our framework's ability to solve various tasks that require learning over long-term episodes, we propose the slow-fast learning loop algorithm, which integrates the TEMP-LORA modules into a dual-speed learning process.

We detail our slow-fast learning loop algorithm in Algorithm 1 (b). Our slow-fast learning loop consists of two primary loops: an inner loop for fast learning and an outer loop for slow learning.

The inner loop, representing the fast learning component, utilizes TEMP-LORA for quick adaptation to each episode. It inherits the fast learning algorithm in Algorithm 1 (a) and updates the memory in the TEMP-LORA throughout the long video generation process. Crucially, the inner loop not only generates long video outputs and updates the TEMP-LORA parameters, but also prepares training data for the slow learning process. It aggregates inputs, ground-truth outputs, and corresponding TEMP-LORA parameters in individual episodes into a dataset $D_s$. The outer loop implements the slow learning process. It leverages the learned frozen TEMP-LORA parameters in different episodes to capture the long-term information of the input data. Specifically, it utilizes the data collected from multiple episodes by the inner loop, which consists of the inputs and ground-truth outputs of each iteration in the episodes, together with the memory saved in TEMP-LORA up to this iteration.

While the slow-fast learning loop may be intensive for full pre-training of the approximate world model, it can be effectively used in finetuning for specific domains or tasks requiring long-horizon planning or experience consolidation. A key application is in robot planning, where slow accumulation of task-solving strategies across environments can enhance the robot's ability to generalize to complex, multi-step challenges in new settings, as illustrated in our experiment (Sec 4.2).

## 3.4 VIDEO PLANNING

We follow Du et al. (2023), formulating task planning as a text-conditioned video generation problem using Unified Predictive Decision Process (UPDP). We define a UPDP as $G = \langle X, C, H, \phi \rangle$, where $X$ is the space of image observations, $C$ denotes textual task descriptions, $H \in \mathcal{T}$ is a finite horizon length, and $\phi(\cdot|x_0, c) : X \times C \to \Delta(X^{T-1})$ is our proposed conditional video generator. $\phi$ synthesizes a video sequence given an initial observation $x_0$ and text description $c$. To transform synthesized videos into executable actions, we use a trajectory-task conditioned policy $\pi(\cdot|x_{t=0}^{T-1}, c) : X^T \times C \to \Delta(A^{T-1})$. Decision-making is reduced to learning $\phi$, which generates future image states based on language instructions. Action execution translates the synthesized video plan $[x_1, ..., x^T]$ to actions. We employ an inverse dynamics model to infer necessary actions for realizing the video plan. This policy determines appropriate actions by taking two consecutive image observations from the synthesized video. For long-horizon planning, ChatGPT decomposes tasks into subgoals, each realized as a UPDP process. After generating a video chunk for a subgoal, we use the inverse dynamics model to execute actions. We sequentially generate video chunks for subgoals, updating TEMP-LORA throughout the long-horizon video generation process.

## 3.5 RETHINKING SLOW-FAST LEARNING IN THE CONTEXT OF COMPLEMENTARY LEARNING SYSTEMS

### *Slow-Fast Learning as in Brain Structures*

In neuroscience, the neocortex is associated with slow learning, while the hippocampus facilitates fast learning and memory formation, thus forming a complementary learning system where the two learning mechanisms complement each other. While slow learning involves gradual knowledge acquisition, fast learning enables rapid formation of new memories from single experiences for quick adaptation to new situations through one-shot contextual learning. However, current pre-training paradigms (*e.g.,* LLMs or diffusion models) primarily emulate slow learning, akin to procedural memory in cognitive science. In our setting, TEMP-LORA serves as an analogy to the hippocampus.

### TEMP-LORA *as Local Learning Rule*

It's long believed that fast learning is achieved by local learning rule (Palm, 2013). Specifically, given pairs of patterns $(x^\mu, y^\mu)$ to be stored in the matrix $C$, the process of storage could be formulated by the following equation:

$$c = \sum_\mu x^\mu y^\mu \tag{4}$$

Consider a sequential memory storage process where learning steps involve adding input-output pairs $(x^\mu, y^\mu)$ to memory, the change $\Delta c_{ij}$ in each memory entry depends only on local input-output interactions (Palm, 2013):

$$\Delta c(t) = x^\mu(t) \cdot y^\mu(t) \tag{5}$$

This local learning rule bears a striking resemblance to LoRA's update mechanism.

$$W' = W + \Delta W = W_{\text{slow}} + W_{\text{fast}} = \Phi + \Theta \tag{6}$$

Where $W_{fast}$ is achieved by the matrix change, updated based on the current-iteration input and output locally as in Equation 3 ($\Delta W \leftarrow z_{0,i-1} \oplus z_{0,i}$).

**Slow-Fast Learning Loop as a Computational Analogue to Hippocampus-Neocortex Interplay**

The relationship between TEMP-LORA and slow learning weights mirrors the interplay between hippocampus and neocortex in complementary learning systems. This involves rapid encoding of new experiences by the hippocampus, followed by gradual integration into neocortical networks (McClelland et al., 1995). As in Klinzing et al. (2019), memory consolidation is the process where hippocampal memories are abstracted and integrated into neocortical structures, forming general knowledge via offline phases, particularly during sleep. This bidirectional interaction allows for both quick adaptation and long-term retention (Kumaran et al., 2016). Our slow-fast learning loop emulates this process, where $W' = W + \Delta W = W_{slow} + W_{fast}$. Here, $W_{fast}$ (TEMP-LORA) rapidly adapts to new experiences, analogous to hippocampal encoding, while $W_{slow}$ gradually incorporates this information, mirroring neocortical consolidation.

## 4 EXPERIMENT

For our experimental evaluation, we will focus on assessing the capabilities of our proposed approach with regard to two perspectives: video generation and video planning. We will detail our baseline models, datasets, evaluation metrics, results, and qualitative examples for each component. Please refer to the supplementary material for experimental setup, implementation details, human evaluation details, computational costs, more ablative studies and more qualitative examples.

### 4.1 EVALUATION ON VIDEO GENERATION

**Baseline Models and Evaluation Metrics**   We compare our SLOWFASTVGEN with several baselines. AVDC (Ko et al., 2023) is a video generation model that uses action descriptions for training video policies in image space. Streaming-T2V (Henschel et al., 2024) is a state-of-the-art text-to-long-video generation model featuring a conditional attention module and video enhancer. We also evaluate Runway (Runway) in an off-the-shelf manner to assess commercial text-to-video generation models. Additionally, AnimateDiff (Guo et al., 2023b) animates personalized T2I models. SEINE (Chen et al., 2023) is a short-to-long video generation method that generates transitions based on text descriptions. iVideoGPT (Wu et al., 2024) is an interactive autoregressive transformer framework. We tune these models (except for Runway) on our proposed dataset for a fair comparison.

We first evaluate the slow learning process, which takes in a previous video together with an input action, and outputs the new video chunk. We include a series of evaluation metrics, including: Fréchet Video Distance (FVD), Peak Signal-to-Noise Ratio (PSNR), Structural Similarity Index (SSIM) and Learned Perceptual Image Patch Similarity (LPIPS). We reserved a portion of our collected dataset as the test set, ensuring no scene overlap with the training set. We evaluate the model's ability to generate consistent long videos through an iterative process. Starting with an initial image and action sequence, the model sequentially generates video chunks by conditioning on the previous chunk and the next action. To assess the consistency of the generated sequence, we use Short-term Content Consistency (SCuts) Henschel et al. (2024), which measures temporal coherence between adjacent frames. We utilize PySceneDetect (PySceneDetect) to detect scene cuts and report their number. We also introduce Scene Revisit Consistency (SRC), a new metric that quantifies how consistently a video represents the same location when revisited via reverse actions (e.g., moving forward and then back). SRC is computed by measuring the cosine similarities between visual features (using ResNet-18 (He et al., 2015)) of the first visit and subsequent visits. We construct an SRC benchmark in Minecraft and Unreal, automating a script to generate specific navigation paths that involve revisiting the same locations multiple times during the course of a long video sequence. Finally, we conduct Human Evaluation, where human raters assess video quality, coherence, and adherence to actions, with each sample rated by at least three individuals on a scale from 0 to 1.

**Result Analysis**   In Table 4, we include the experimental results of video generation, where the left part focuses on slow learning for action-conditioned video generation, and the right part evaluates the ability to generate consistent long videos. From the table, we can see that our model outperforms others with regard to both slow learning and long video generation. Specifically, our method achieves a notable FVD score of 514, significantly lower than those of other methods, such as AVDC (1408) and Runway Gen-3 Turbo (1763), demonstrating its capability to capture the underlying dynamics of video content. The high PSNR and SSIM scores also indicate that our model achieves high visual fidelity and similarity with ground-truth videos. Runway produces less satis-

| | FVD ↓ | PSNR ↑ | SSIM ↑ | LPIPS ↓ | SCuts ↓ | SRC ↑ | Human |
|---|---|---|---|---|---|---|---|
| AVDC | 1408 | 16.96 | 52.63 | **20.65** | 3.13 | 83.89 | 0.478 |
| Streaming-T2V | 990 | 14.87 | 48.33 | 33.00 | 0.89 | 91.02 | 0.814 |
| Runway Gen-3 Turbo | 1763 | 11.15 | 47.29 | 52.71 | 2.46 | 80.26 | 0.205 |
| AnimateDiff | 782 | 17.89 | 52.34 | 33.41 | 2.94 | 90.12 | 0.872 |
| SEINE | 919 | 18.04 | 54.15 | 35.72 | 1.03 | 88.95 | 0.843 |
| iVideoGPT | 1303 | 13.08 | 31.37 | 27.22 | 1.32 | 82.19 | 0.536 |
| Ours (wo/ Temp-LoRA) | / | / | / | / | 1.88 | 89.04 | 0.869 |
| **Ours** SLOWFAST-VGEN | **514** | **19.21** | **60.53** | 25.06 | **0.37** | **93.71** | **0.897** |

Table 1: **Video Generation Results**. SLOWFAST-VGEN outperforms baselines both in slow learning and long video generation, while achieving good scores in human evaluation.

factory results and could not comprehend text inputs, suggesting that state-of-the-art commercial models still struggle to serve as action-conditioned world models. From the long video generation perspective, we can see that our model and Streaming-T2V outperform other models by a large margin, with fewer scene changes (indicated by SCuts) and better scene revisit consistency. We attribute the notable improvement in performance to the mechanisms of the two models specially designed to handle short- and long-term memory. However, Streaming-T2V is still inferior to our model, since the appearance preservation module it uses only takes in one anchor frame, neglecting the full episode memory along the generation process. On the other hand, our TEMP-LORA module takes into account the full trajectory, therefore achieving better consistency. We also observe that AVDC suffers from incoherence and ambiguity in output videos, which might be blamed to the image-space diffusion, which results in instability and low quality compared with latent-space diffusion models.

**Qualitative Study** Figure 3 shows the results of slow learning. Our model is able to encompass a variety of scenarios in action-conditioned video generation, including driving scenes, robotics, human avatars, droid scenes *etc*. The model is able to condition on and conform to various actions (*e.g.*, in the third row, it manages to go in different directions given different prompts). We also observe that our model adheres to physical constraints. For instance, in the last action for row three, it knows to come to a halt and bring the hands to a resting position. In Figure 4, we compare our model's results with and without the TEMP-LORA module. The TEMP-LORA improves quality and consistency in long video generation, demonstrating more smoothness across frames. In the first example, the scene generated without TEMP-LORA begins to morph at t = 96, and becomes completely distorted and irrelevant at t = 864. The scene generated with TEMP-LORA almost remains unchanged for the first 400 frames, exhibiting only minor inconsistencies by t = 896. Our model is able to generate up to 1000 frames without significant distortion and degradation. Additionally, the model lacking TEMP-LORA suffers from severe hallucinations for the second example, such as generating an extra bread at t = 48, changing the apple's color at t = 144, and adding a yellow cup at t = 168. Although the robotics data does not involve long-term videos with sequential instructions, but rather short episodes of single executions, our model still performs well by conditioning on previous frames. With TEMP-LORA, our model also produces less noise in later frames (*e.g.*, t = 216). As the scene is unseen in training, our model also demonstrates strong adaptation ability.

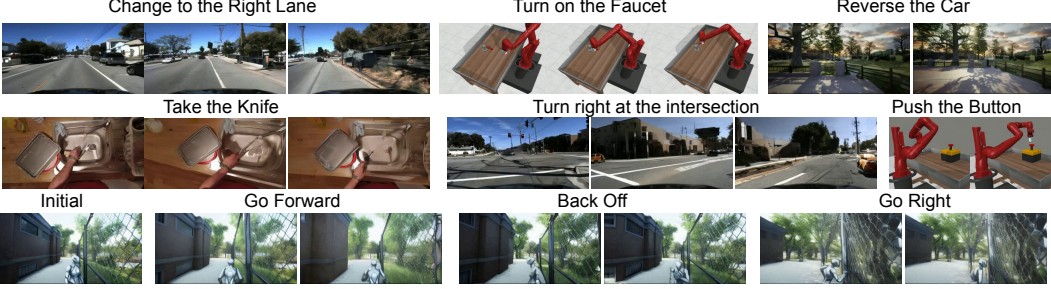

Figure 3: **Qualitative Examples of Video Generation with Regard to Slow Learning**. Our SLOWFAST-VGEN is able to conduct action-conditioned video generation in diverse scenarios.

## 4.2 EVALUATION ON LONG-HORIZON PLANNING

In this section, we show how SLOWFAST-VGEN could benefit long-horizon planning. We carefully design two tasks which emphasize the memorization of previous trajectories, in the domains of robot manipulation and game navigation respectively. We follow the steps in Section 4.2 for task execution and employ the slow-fast learning loop for these two specific tasks separately. We report the distance to pre-defined waypoints as well as the FVD of the generated long videos.

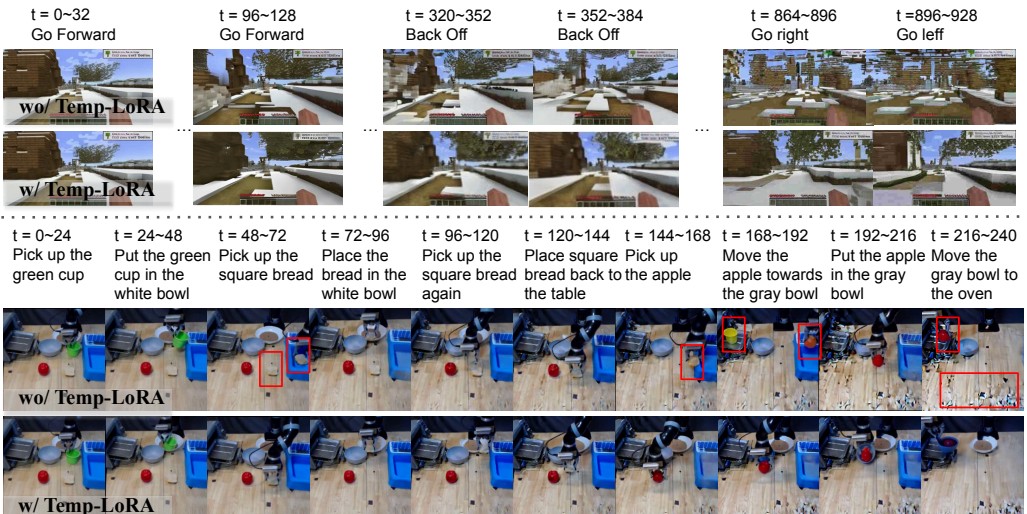

Figure 4: **Qualitative Examples of Fast Learning for Video Generation**. t means frame number. Red boxes denote objects inconsistent with before. TEMP-LoRA boosts consistency in long videos.

| | AVDC | | Streaming-T2V | | Ours wo Temp-LoRA | | Ours wo Loop | | Ours | |
|---|---|---|---|---|---|---|---|---|---|---|
| | Dist | FVD | Dist | FVD | Dist | FVD | Dist | FVD | Dist | FVD |
| RLBench | 0.078 | 81.4 | 0.022 | 68.2 | 0.080 | 70.3 | 0.055 | 69.8 | **0.013** | **65.9** |
| Minecraft | 5.57 | 526 | 2.18 | 497 | 6.31 | 513 | 2.23 | 501 | **1.51** | **446** |

Table 2: **Long-horizon Planning Results.** Our model outperforms baselines in both tasks.

**Robot Manipulation** We build a new task from scratch in the RLBench (James et al., 2019) environment. The task focuses on moving objects and then returning them to previous locations. We record the distance to the previous locations of the cubes. Table 4.2 shows that our model and Streaming-T2V outperform models without memories. Streaming-T2V has satisfying results since the appearance preservation module takes an anchor image as input, yet is still inferior to our model. Ablative results of "Ours wo Loop" also demonstrate the importance of the slow-fast learning loop.

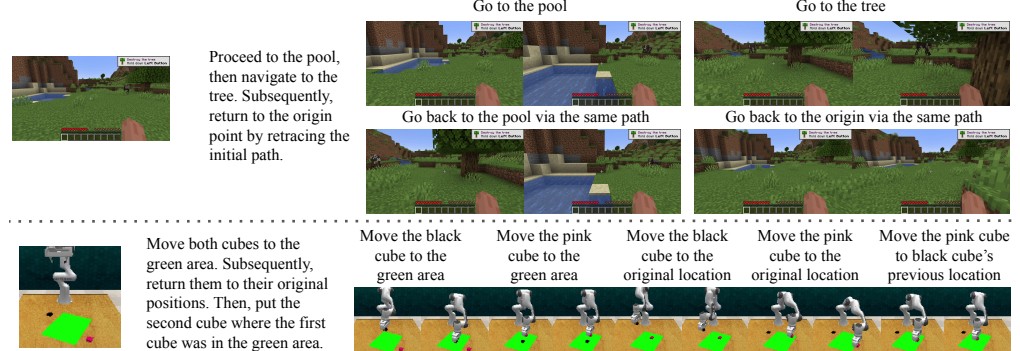

Figure 5: **Qualitative Example of Planning.** SLOWFAST-VGEN can retain long-term memory.

**Game Navigation** We develop a task in Minecraft that requires the gamer to retrace the initial path to return to a point. We define a set of waypoints along the way, and measure the closest distance to these waypoints. From Table 4.2 and Figure 5, we can see that our model achieves superior results.

## 5 CONCLUSION AND LIMITATIONS

In this paper, we present SLOWFAST-VGEN, a dual-speed learning system for action-driven long video generation. Our approach combines slow learning for building comprehensive world knowledge with fast learning for rapid adaptation and maintaining consistency in extended videos. Through experiments, we demonstrate SLOWFAST-VGEN's capability to generate coherent, action-responsive videos and perform long-horizon planning. One limitation is that our model could not suffice as an almighty world model as many scenarios are not covered in the dataset. Our method also introduces additional computation during inference as specified in the Supplementary Material.

## ACKNOWLEDGEMENT

The work was done when Yining Hong was an intern at Microsoft Research. The work was also partially supported by NSF DMS-2015577, NSF DMS-2415226, and a gift fund from Amazon. We would like to thank Yan Wang and Dongyang Ma for insights into TEMP-LORA for text generation. We would also like to thank Siyuan Zhou, Xiaowen Qiu, Xingcheng Yao and Johnson Lin for insights into video diffusion models. We would like to thank PLUSLab paper clinic for proofreading.

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

article iclr2025$_c on ference, times$

amsmath,amsfonts,bm

OT1phvmsl

boldOT1phvbxn

*arg max $arg\,max$ ∗ arg min$arg\,min$

sign $sign$Tr$Tr$

hyperref url algorithm algorithmic graphicx wrapfig xcolor enumitem colortbl

## CONTENTS

## A    CONTRIBUTION STATEMENT

**Yining Hong** was responsible for all of the code development, paper writing, and experiments. She also collected the data for Minecraft.

**Beide Liu** contributed to most of the data collection with regard to Unreal data. He was in charge of setting up the Unreal Engine, purchasing assets online, writing the Python scripts for automate agent control, and recording first-person and third-person videos of Unreal data.

**Maxine Wu** collected the data of Google 3D Tiles. She was also responsible for the task setup of RLBench and the data collection of RLBench. She also curated part of the driving data.

**Yuanhao Zhai** wrote the codes for AnimateDiff, which was one of the baseline models.

The other people took on the advising roles, contributing extensively to the project by offering innovative ideas, providing detailed technical recommendations, assisting with troubleshooting code issues, and conducting multiple rounds of thorough paper reviews. They provided valuable expertise on video diffusion models. **Zhengyuan Yang, Yingnian Wu and Lijuan Wang** were involved in brainstorming and critical review throughout the project. Specifically, **Zhengyuan Yang** provided much technical support. **Yingnian Wu** came up with the idea of TEMP-LORA for modelling episodic memory as well as the masked video diffusion model. **Lijuan Wang** provided valuable insights throughout the project.

## B    MORE DETAILS ABOUT THE METHOD

### B.1    PRELIMINARIES ON LATENT DIFFUSION MODELS

Stable Diffusion (Rombach et al., 2022), operates in the compressed latent space of an autoencoder obtained by a pre-trained VAE . Given an input $x_0$, the process begins by encoding it into a latent representation: $z_0 = E(x_0)$ where $E$ is the VAE encoder function. Noise is then progressively added to the latent codes through a Gaussian diffusion process:

$$q(z_t|z_{t-1}) = \mathcal{N}(z_t; \sqrt{1 - \beta_t} z_{t-1}, \beta_t \mathbf{I}) \tag{7}$$

for $t = 1, \ldots, T$, where $T$ is the total number of diffusion steps and $\beta_t$ are noise schedule parameters. This iterative process can be expressed in a simpler form:

$$z_t = \sqrt{\bar{\alpha}_t} z_0 + \sqrt{1 - \bar{\alpha}_t} \epsilon \tag{8}$$

where $\epsilon \sim \mathcal{N}(0, \mathbf{I})$, $\bar{\alpha}_t = \prod_{i=1}^{t} \alpha_i$, and $\alpha_i = 1 - \beta_i$. Stable Diffusion employs an $\epsilon$-prediction approach, training a neural network $\epsilon_\theta$ to predict the noise added to the latent representation. The loss function is defined as:

$$L = \mathbb{E}_{t, z_0 \sim p_{\text{data}}, \epsilon \sim \mathcal{N}(0,1), c} \left[ ||\epsilon - \epsilon_\theta(z_t, t, c)||_2^2 \right] \tag{9}$$

Here, $c$ represents the conditioning (e.g., text), and $\theta$ denotes the neural network parameters, typically implemented as a U-Net (Ronneberger et al., 2015).

During inference, the model iteratively denoises random Gaussian noise, guided by the learned $\epsilon_\theta$, to generate latent representations. These are then decoded to produce high-quality images consistent with the given textual descriptions.

Video diffusion models (Ho et al., 2022b) typically build upon LDMs by utilizing a 3D U-Net architecture, which enhances the standard 2D structure by adding temporal convolutions after each spatial convolution and temporal attention blocks following spatial attention blocks.

### B.2    PRELIMINARIES ON LOW-RANK ADAPTATION (LORA)

LoRA Hu et al. (2021) transforms the fine-tuning process for large-scale models by avoiding the need to adjust all parameters. Instead, it utilizes compact, low-rank matrices to modify only a subset of the model's weights. This approach keeps the original model parameters fixed, addressing the problem of catastrophic forgetting, where new learning can overwrite existing knowledge. LoRA

utilizes compact, low-rank matrices to modify only a subset of the model's weights, therefore avoiding the need to adjust all parameters. In LoRA, the weight matrix $W \in \mathbb{R}^{m \times n}$ is updated by adding a learnable residual. The modified weight matrix $W'$ is:

$$W' = W + \Delta W = W + AB^T$$

where $A \in \mathbb{R}^{m \times r}$ and $B \in \mathbb{R}^{n \times r}$ are low-rank matrices, and $r$ is the rank parameter that determines their size. In this paper, we denote the LoRA finetuning as the fast learning process and the pre-training as slow learning process. The equation then becomes:

$$W' = W + \Delta W = W_{\text{slow}} + W_{\text{fast}} = \Phi + \Theta \tag{10}$$

where $\Phi$ corresponds to the pre-trained slow-learning weights, and $\Theta$ corresponds to the LoRA paraemters in the fast learning phase.

### B.3    MODELSCOPET2V DETAILS

We base our slow learning model on ModelscopeT2V (Wang et al., 2023). Here, we introduce the details of this model.

Given a text prompt $p$, the model generates a video $v_{pr}$ through a latent video diffusion model that aligns with the semantic meaning of the prompt. The architecture is composed of a visual space where the training video $v_{gt}$ and generated video $v_{pr}$ reside, while the diffusion process and denoising UNet $\epsilon_\theta$ operate in a latent space. Utilizing VQGAN, which facilitates data conversion between visual and latent spaces, the model encodes a training video $v_{gt} = [f_1, \ldots, f_F]$ into its latent representation $Z_{gt} = [E(f_1), \ldots, E(f_F)]$. During the training phase, the diffusion process introduces Gaussian noise to the latent variable, ultimately allowing the model to predict and denoise these latent representations during inference.

To ensure that ModelScopeT2V generates videos that adhere to given text prompts, it incorporates a text conditioning mechanism that effectively injects textual information into the generative process. Inspired by Stable Diffusion, the model augments the UNet structure with a cross-attention mechanism that allows for conditioning of visual content based on textual input. The text embedding $c$ derived from the prompt $p$ is utilized as the key and value in the multi-head attention layer, enabling the intermediate UNet features to integrate text features. The text encoder from the pre-trained CLIP ViT-H/14 converts the prompt into a text embedding, ensuring a strong alignment between language and vision embeddings.

The core of the latent video diffusion model lies in the denoising UNet, which encompasses various blocks, including the initial block, downsampling block, spatio-temporal block, and upsampling block. Most of the model's parameters are concentrated in the denoising UNet $\epsilon_\theta$, which is tasked with the diffusion process in the latent space. The model aims to minimize the discrepancy between the predicted noise and the ground-truth noise, thereby achieving effective video synthesis through denoising. ModelScopeT2V's architecture also includes a spatio-temporal block, which captures complex spatial and temporal dependencies to enhance video synthesis quality. The spatio-temporal block is comprised of spatial convolutions, temporal convolutions, and attention mechanisms. By effectively synthesizing videos through this structure, ModelScopeT2V learns comprehensive spatio-temporal representations, allowing it to generate high-quality videos. The model implements a combination of self-attention and cross-attention mechanisms, facilitating both cross-modal interactions and spatial modeling to capture correlations across frames effectively.

## C    DATASET STATISTICS

We provide the dataset statistics in Figure 6.

## D    MORE EXPERIMENTAL DETAILS

### D.1    EXPERIMENTAL SETUP AND IMPLEMENTATION DETAILS

We utilize approximately 64 V100 GPUs for the pre-training of SLOWFAST-VGEN, with a batch size of 128. The slow learning rate is set to 5e-6, while the fast learning rate is 1e-4. Training

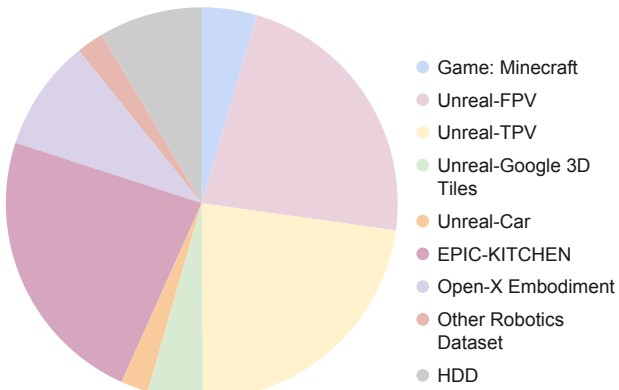

Figure 6: **Statistics of our Training Dataset.**

videos of mixed lengths are used, all within the context window of 32 frames. During training, we freeze the VAE and CLIP Encoder, allowing only the UNet to be trained. For inference and fast learning, we employ a single V100 GPU. For TEMP-LORA, a LoRA rank of 32 is used, and the Adam optimizer is employed in both learning phases.

## D.2 COMPUTATION COSTS

In Table 3, we show the computation costs with and without TEMP-LORA. While the inclusion of TEMP-LORA does introduce some additional computation during the inference process, the difference is relatively minor and remains within acceptable limit.

|  | Ours wo TEMP-LORA | Ours w TEMP-LORA |
|---|---|---|
| Average Inference Time per Sample (seconds) | 12.9305 | 13.8066 |
| Inference Memory Usage (MB) | 9579 | 9931 |

Table 3: **Comparison of Computation Costs with and without TEMP-LORA**

## D.3 HUMAN EVALUATION DETAILS

In our human evaluation session for action-conditioned long video generation, 30 participants assessed the generated video samples (50 videos per person) based on three criteria:

- Video Quality (0 to 1): Participants evaluated the overall visual quality, considering aspects such as resolution, clarity, and aesthetic appeal.
- Coherence (0 to 1): They examined the logical flow of actions and whether the events progressed seamlessly throughout the video, ensuring there were no abrupt changes or disconnections.
- Adherence to Actions (0 to 1): Participants judged how accurately the generated videos reflected the specified action prompts, assessing whether the actions were effectively depicted.

Each video was rated by at least three different individuals to ensure reliability. The collected ratings were then compiled for analysis, with average scores calculated to assess performance across the different criteria.

## E  EXPERIMENTS ON ABLATIONS AND VARIATIONS OF SLOWFAST-VGEN

We introduce several variations of SLOWFAST-VGEN, including:

|  | SCuts ↓ | SRC ↑ |
|---|---|---|
| Our (w original TEMP-LORA) | 0.55 | 92.24 |
| Ours (wo Local Learning Rule) | **0.36** | 90.27 |
| Ours (wo Chunk Input) | 1.24 | 90.01 |
| Ours (wo/ Temp-LoRA) | 1.88 | 89.04 |
| **Ours SLOWFAST-VGEN** | 0.37 | **93.71** |

Table 4: **Scene Cuts and SRC Scores**. Comparison of scene cuts and SRC scores for our method with and without Temp-LoRA.

- **Ours wo Chunk Input** that only conditions on single-frame images instead of previous chunk
- **Ours wo Local Learning Rule** that samples over the whole generated sequence for training TEMP-LORA, instead of using local inputs and outputs to train.
- **Ours w original TEMP-LORA** that uses the original TEMP-LORA structure that were designed for long text generation.

We show the results below. From the table, we can see that SLOWFAST-VGEN trained over sampled full sequence also shows good performances. However, our observation indicates that this method tends to over-smooth the generated sequences, leading to blurry videos for later frames.

# F    MORE QUALITATIVE EXAMPLES

## F.1    MORE QUALITATIVE EXAMPLES OF SLOW LEARNING

In Figure 7 and Figure 8, we include more qualitative examples with regard to slow learning.

## F.2    MORE QUALITATIVE EXAMPLES OF FAST LEARNING

In Figure 9 and Figure 10, we include more qualitative examples with regard to fast learning.

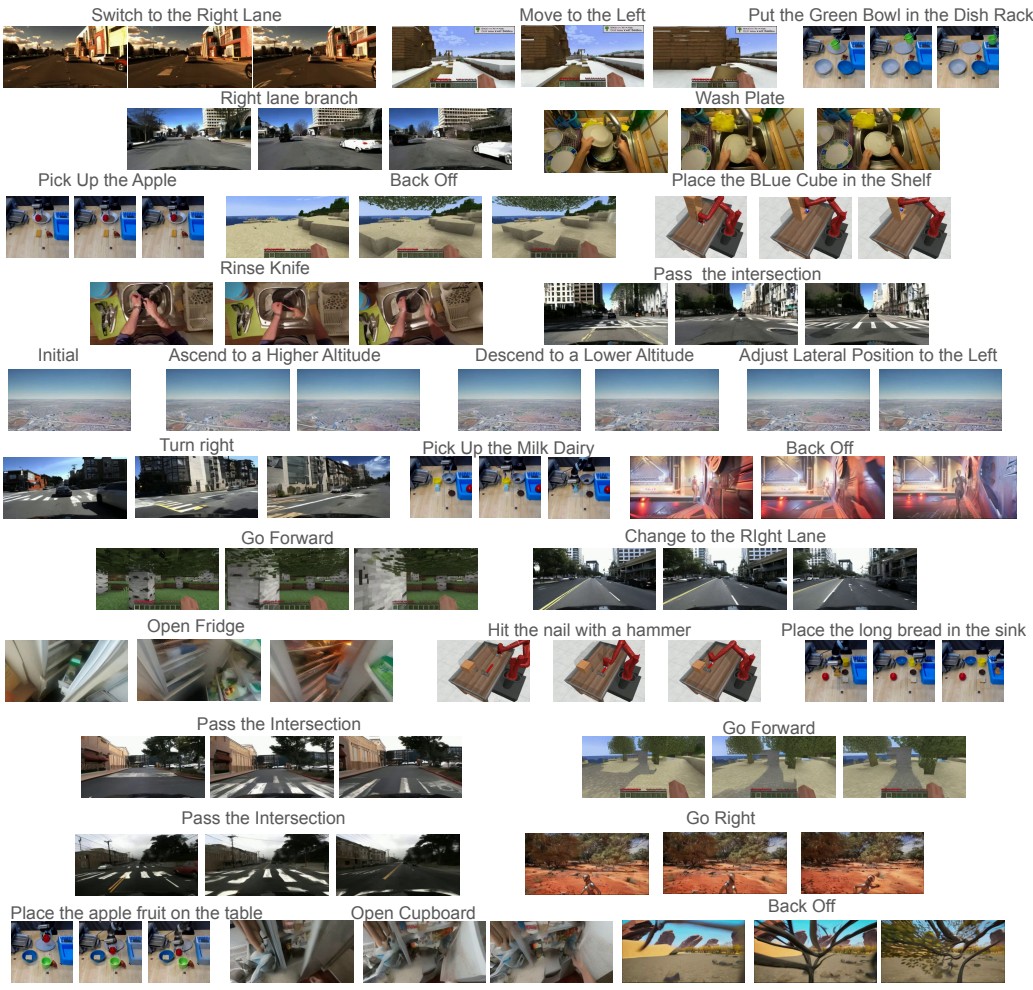

Figure 7: Qualitative Examples on Slow Learning. Part 1.

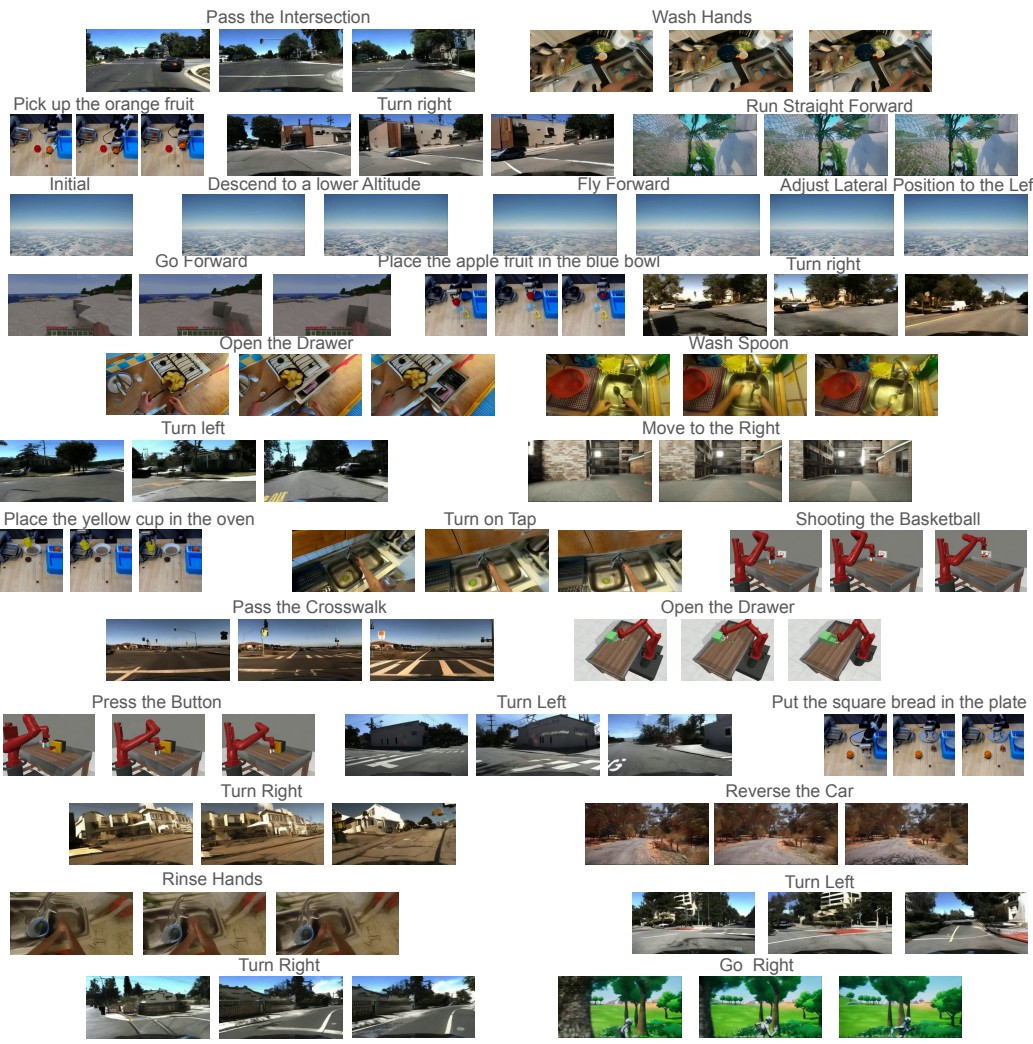

Figure 8: Qualitative Examples on Slow Learning. Part 2.

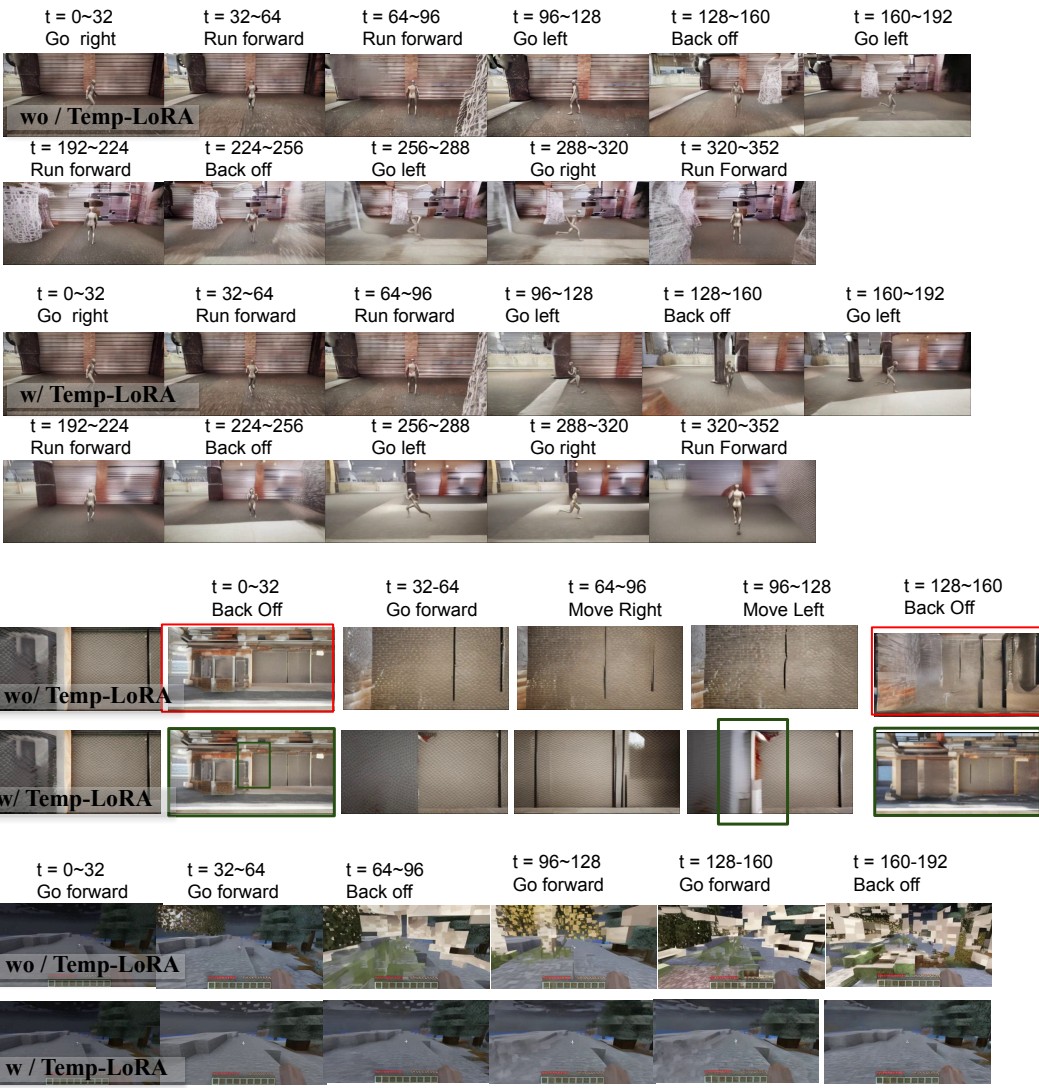

Figure 9: Qualitative Examples on Fast Learning. Part 1. We mark consistent objects / frames in green bounding boxes and inconsistent ones in red.

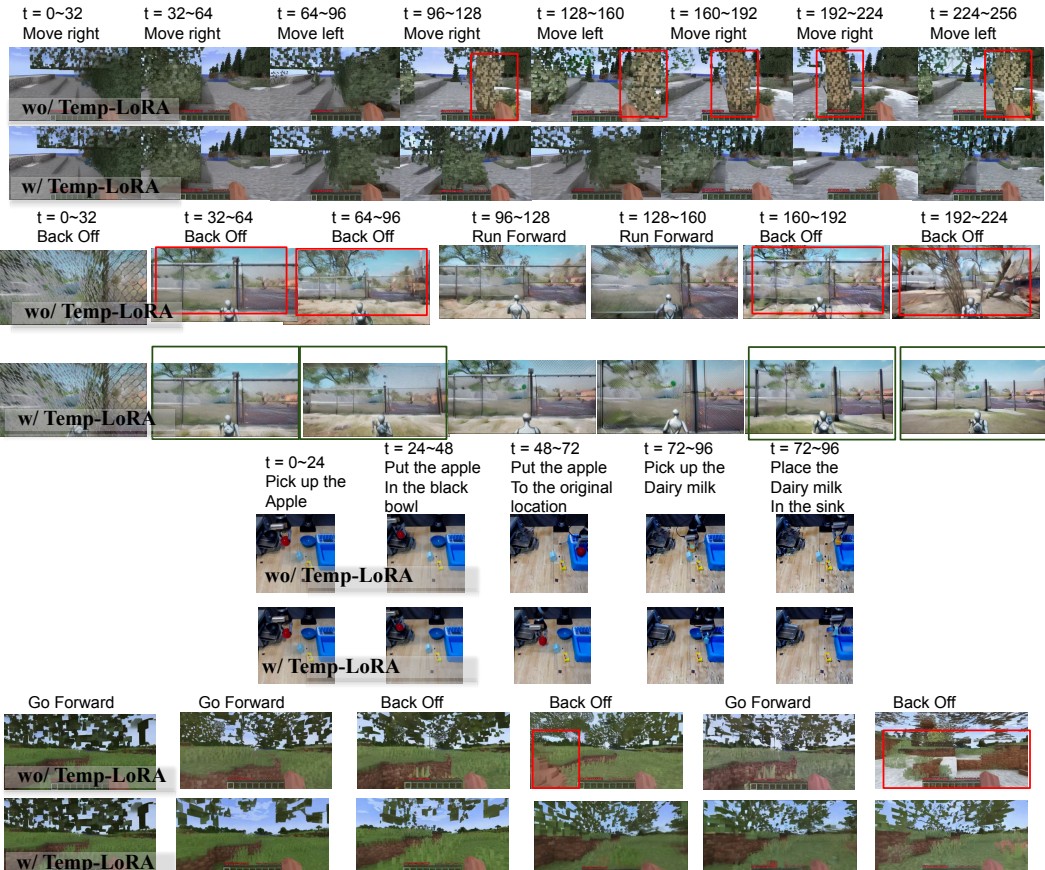

Figure 10: Qualitative Examples on Fast Learning. Part 2. We mark consistent objects / frames in green bounding boxes and inconsistent ones in red.

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
