# OpenReview forum: "SlowFast-VGen: Slow-Fast Learning for Action-Driven Long Video Generation"
_ICLR.cc/2025/Conference — ICLR 2025 Spotlight_

### Official Review · Reviewer_bXBH · 2024-10-20

**Soundness:** 3
**Presentation:** 3
**Contribution:** 3
**Rating:** 8
**Confidence:** 4

**Summary:**

This paper proposes a slow-fast structure for retaining episodic memory in long video generation. The “slow” video model follows similarly to prior works in masked conditional video diffusion models, where x prior frames are conditioned on to generate y future frames. To incorporate longer-context memory when generating multiple chunks in a row, the authors introduce temporal LoRA parameters that are updated during inference time to store episodic memory. For more general long-context generation tasks, the paper additionally proposes a slow-fast training algorithm.

**Strengths:**

* The paper is generally easy to follow, and well written
* The proposed algorithm is interesting, and could be a promising direction for incorporating memory through test-time training in neural networks
* Experimental results on long videos show better consistency than baseline methods, and without using their temporal LoRA module.
* The authors introduce a collection of action-conditioned video that could be useful for future related research

**Weaknesses:**

* The purpose of the “slow” video baseline comparisons are a little confusing to me. Is this not just showing that training on specifically curated action data results in a model that can better generate conditioned on actions?
* The authors have shown some benefits in retaining long-term memory over the course of multi-chunk video generation. However, I am still a little doubtful at the long-context abilities of this algorithm if used at scale. It seems difficult to propagate / learn abstract representations in long term memory that may be useful for long-term video generation, especially since the temporal LoRA is updated using a pixel-wise diffusion loss and thus biased to memorizing a specific instance of a scene. How would this model perform in more difficult long context tasks (e.g. remembering something seen 10-30s ago, or trying to localize position / orientation in 3D space), such as the datasets introduced in [1]? Or, in a more simpler environment, making one full rotation, and then randomly turning left and right (the first rotation should newly generate the scene, and afterwards only requires accurate scene retrieval from memory).


[1] Yan, Wilson, et al. "Temporally consistent transformers for video generation." International Conference on Machine Learning. PMLR, 2023.

**Questions:**

* It would be useful if the authors could include some way to view the original video samples (not just snapshots of frames in the paper). Perhaps with an anonymous site?
* How long are the generated videos in seconds? Most figures show timestamps, but framerate is not mentioned in the paper.

---

> ### Author Response · Authors · 2024-11-20
> **Response to Reviewer bXBH (1/3)**
>
> *We are extremely grateful for your thorough and insightful feedback, which has significantly enhanced the quality of our paper! Following your valuable suggestions, we have conducted additional experiments and revised our manuscript accordingly to address all your comments.*
>
> &nbsp;
>
> > **W1: The purpose of the “slow” video baseline comparisons are a little confusing to me. Is this not just showing that training on specifically curated action data results in a model that can better generate conditioned on actions?**
>
> We appreciate this question! We apologize for lack of emphasis on slow learning, which results in your concern! We would like to clarify two key points:
>
> * As stated in Line 407 in the paper, all baseline models except for Runway (including AVDC, Streaming-T2V, etc.) are tuned using our action-conditioned dataset. Therefore, the performance difference cannot be attributed solely to the training data.
> * Beyond the dataset, we also propose a specific architecture for slow learning based on masked conditional video diffusion. Our experiments demonstrate that this architecture is more effective than other approaches (e.g., pixel-space diffusion without using latents, adding channels for conditioning instead of using masked diffusion) for action-conditioned generation, as evidenced by better FVD scores (514 vs >780) and visual quality across different scenarios.
>
> We appreciate the reviewer's insight and acknowledge that we did not sufficiently emphasize the importance of slow learning in our paper. We have added one line discussing the slow learning's critical role in temporal modeling to Section 3.2. Hope this addition addresses your concern!

---

> ### Author Response · Authors · 2024-11-20
> **Response to Reviewer bXBH (2/3)**
>
> > **W2: The authors have shown some benefits in retaining long-term memory over the course of multi-chunk video generation. However, I am still a little doubtful at the long-context abilities of this algorithm if used at scale. It seems difficult to propagate / learn abstract representations in long term memory that may be useful for long-term video generation, especially since the temporal LoRA is updated using a pixel-wise diffusion loss and thus biased to memorizing a specific instance of a scene. How would this model perform in more difficult long context tasks (e.g. remembering something seen 10-30s ago, or trying to localize position / orientation in 3D space), such as the datasets introduced in [1]? Or, in a more simpler environment, making one full rotation, and then randomly turning left and right (the first rotation should newly generate the scene, and afterwards only requires accurate scene retrieval from memory).**
>
> Thank you for raising this concern and introducing such a useful dataset [1]! We have cited this paper and carried out additional experiments on this dataset.
>
> * **Current Long-context Evidence**
> We acknowledge that our current pre-training dataset involves more scenes with position changes rather than camera rotations. However, we would like to point out that these scenes can equally demonstrate our model's ability to retain long-context memory, as can be seen from the qualitative examples in our supplementary material Section E.2 and in our [visualization website for rebuttal](https://sites.google.com/view/slowfastvgen/home). Thank you for the inspiring suggestion and we will include more "rotation" scenes in the future!
>
> * **Extentive Experiments on Teco**
> We would like to thank the reviewer so much for pointing out this awesome dataset (Teco[1])! To further resolve the reviewer's concerns, we finetune our model on the Teco-Habitat dataset. We apologize that due to time and resource limit, we were only able to train on 10% of these data for only 10,000 iterations with a batch size 64. We show the quantitative results below as well as in the revise Appendix Section D.7, and qualitative results of action-conditioned long video generation in [the visualization website for rebuttal](https://sites.google.com/view/slowfastvgen/home) Section 7.
>
> &nbsp;
>
> **Table A. Extensive Experiments on Teco-Habitat**
> | Method | SCuts $\downarrow$ | SRC $\uparrow$ |
> |--------|-------------------|----------------|
> | AVDC | 15.47 | 75.21 |
> | Streaming-T2V | 8.92 | 82.45 |
> | Runway Gen-3 Turbo | 13.85 | 71.34 |
> | AnimateDiff | 14.23 | 81.56 |
> | SEINE | 9.87 | 80.12 |
> | iVideoGPT | 11.45 | 73.88 |
> | Ours (wo/ Temp-LoRA) | 12.64 | 79.92 |
> | **Ours SlowFast-VGen** | **1.12** | **85.43** |
>
> &nbsp;
>
> &emsp;As the original setting for Teco, we use the first 144 frames as our conditioned frames. Specifically, we update our Temp-LoRA parameters using the first 144 frames to store the memory, and calculate results on the remaining frames. Our qualitative analysis demonstrates that our model maintains significantly better temporal coherence and visual fidelity compared to the variant without Temp-LoRA, particularly during extended generation sequences (60+ action-conditioned iterations). While the baseline model without Temp-LoRA exhibits rapid degradation and noise artifacts in early generation phases, our full model preserves consistent visual quality and temporal relationships throughout the sequence.
>
> &emsp;We sincerely apologize for not being able to provide complete results on Teco. Due to time constraints during the rebuttal period, we were only able to conduct preliminary experiments on 10% data for very limited iterations. These initial results demonstrate promising performance of our model. We commit to including comprehensive experiments on Teco in the camera-ready version. We appreciate the reviewer's understanding on this matter, and we would like to thank the reviewer again for introducing this wonderful dataset.
>
> * **Technical Clarification**
> We want to clarify that our model doesn't operate at the pixel level - it works in the compressed latent space of a pre-trained VAE. The diffusion process and loss are computed in this latent space, enabling more abstract representations to be saved in the LoRA parameters and more efficient memory storage. We're sorry that we did not make this clear enough in the paper, and will revise the paper for better in the camera-ready version. We want to thank the reviewer for raising this concern.
>
> [1] Yan, Wilson, et al. "Temporally consistent transformers for video generation." International Conference on Machine Learning. PMLR, 2023.

---

> ### Author Response · Authors · 2024-11-20
> **Response to Reviewer bXBH (3/3)**
>
> > **Q1: It would be useful if the authors could include some way to view the original video samples (not just snapshots of frames in the paper). Perhaps with an anonymous site?**
>
> Thank you for the suggestion! The anonymous site is at https://sites.google.com/view/slowfastvgen/home.
>
> &nbsp;
>
> > **Q2: How long are the generated videos in seconds? Most figures show timestamps, but framerate is not mentioned in the paper.**
>
> We apologize that we did not clarify this enough in our original submission.
> * Note that in our paper, t=32 means the 32th frame, as specified in the captions of the figures (*e.g.,* Line 503.)
> * We use an fps of 4, which results in 250 seconds of videos in maximum.
>
> &nbsp;
>
> *We believe these comprehensive revisions and additional experiments have substantially strengthened our paper and addressed your thoughtful comments. We hope that the improved manuscript better demonstrates the technical merits and broader impact of our work. If you have further questions, please don't hesitate to let us know in the rebuttal window! Thank you again for your invaluable feedback in helping us enhance this paper!*
>
> &nbsp;
>
> Best,
> Authors

---

> > ### Comment · Reviewer_bXBH · 2024-11-20
> > **Response**
> >
> > Thank you to the authors for the detailed response.
> >
> > > Long context capabilities
> >
> > I appreciate the authors for trying to run a simple experiment on the teco dataset, however I’m still not very convinced that the model would scale well to learning more difficult long-term dependencies in video. Fundamentally, it is not unclear to me how the LoRA would be able to learn these long-term abstract representations (e.g. internally representing a 3D map of a scene, tracking position / orientation, or tracking the ongoing narrative of a movie), even when scaled to a large amount of compute, more data, etc.
> >
> > Most experiments show long term dependencies through forward / backward motion, which still is a fairly simple task and generally only requires memorizing snapshots of past frames seen, which the LoRA can be good at. Loss is applied in a latent space, as opposed to pixel space, but VAE representations are still very image-like / spatially biased (i.e. one can rescale + resize the VAE latent and visualize it and it is similar corrupted version of the original image. The same observation applies to video VAEs that downsample spatio-temporally).
> >
> > I think the paper would be much stronger if the authors could demonstrate more complex long-term understanding that does not require just memorization. For example, if the model was able to synthesize historical context to accurately generate a room when coming in from a different door / viewpoint it has not seen yet.
> >
> > **However, I also think it’s okay if this is out of scope of the current work. Test-time training for long video generation is generally an under-studied area, yet a potentially exciting direction. This paper presents a step forward in that direction, and could be of interest to others in the field, so I will raise my score to an accept, as most of my other concerns have been addressed.**

---

> > > ### Author Response · Authors · 2024-11-21
> > > **Further Response to Reviewer bXBH**
> > >
> > > We sincerely thank you for your profound insights into video generation. We are deeply honored to receive guidance from someone who not only reads our paper with such careful attention and truly understands our approach, but also demonstrates encyclopedic knowledge in this field while raising crucial directions for improvement.
> > >
> > > We did start this project with the goal to maintain an abstract 3D "cognitive map" of the environment - this "cognitive map" may assist in navigation tasks and drive the agent's policy at obstacles and landmarks. The first framework we came up with was to explicitly construct the cognitive map using techniques like Dust3R and encode this 3D representation. However, this framework seems complex and engineering-driven.
> > >
> > > On the contrary, we always believe that a good abstraction of a video sequence will give us objects, 3D, physics, and can be represented by sparse latents of these perspectives. **We believe there are two levels of abstractions in our framework: the latent space of the diffusion model, and the LoRA parameters themselves.** While the latent space of current video diffusion models might still be relatively image-like ("similar corrupted version of the original image"), the LoRA parameters potentially serve as a higher-level abstraction that could encode more structural information including 3D geometry, object relationships, and physics. Our current TEMP-LoRA implementation demonstrates this through its ability to maintain temporal consistency via abstract representations rather than simple frame snapshots - as evidenced by its ability to remember scene elements like barrels in previously visited locations (examples on our website). However, we acknowledge that currently TEMP-LoRA does not give us satisfying results towards true 3D cognitive mapping that's as good as achieved by direct 3D reconstruction. We believe future improvements should focus on making these representations more **conceptually dense and structurally informative** for better data efficiency and generalization.
> > >
> > > The distinction between memory types in cognitive science is also relevant here - our model currently excels at episodic memory (temporal sequences and events) but may not yet fully capture sophisticated allocentric spatial representations. This aligns with theories in neuroscience suggesting that spatial and temporal representations might be processed through partially distinct mechanisms. Additionally, the place cells in the hippocampus (fast learning) are responsible for spatial encoding, but the consolidation and semantic mapping happens at the Neocortex (slow learning).
> > >
> > > Given this perspective, we believe our slow-fast learning loop has strong potential for cognitive mapping, particularly through the LoRA parameters serving as abstract representations. While we've implemented slow learning + fast learning on Teco-Habitat, we haven't yet explored the full slow-fast learning loop. Theoretically, the slow-learning process could utilize fast learning parameters (which encode individual habitat scenes) to develop general skills for 3D mapping and navigation.
> > >
> > > We are motivated to try slow-fast learning loop for Teco scenes and hope to integrate these results in our camera-ready version (we hope we can provide results in the rebuttal period but we are unsure how long it will take to have preliminary results).
> > >
> > > Thanks again!

---

### Official Review · Reviewer_iXgc · 2024-10-31

**Soundness:** 3
**Presentation:** 3
**Contribution:** 3
**Rating:** 8
**Confidence:** 4

**Summary:**

In this paper, the authors tackle the challenging action-driven video generation task. They approach it by dividing the model and training into a slow and fast learning loop, reflecting the changes in the  dynamics in the world model and in individual episodes. This is achieved by adding a temporal LoRa to the base architecture, which is also used during inference. While minimally increasing the inference time, this temporal-LoRa leads to an improved generative performance and temporal consistency.
To train the model the authors create a new large scale video-action-annotation dataset. The model performance is evaluated on video generation tasks and long-horizon planning.

**Strengths:**

1. The paper is well structured, and additional details in the appendix help to clarify some aspects.
2. The authors introduce a bio-inspired interpretation of a slow-and fast learning loop to improve long video generation by maintining an episodic memory, and link the inspiration to the hippocampus-cortex learning hypothesis. I like the independence of the lora loop from the action label embedding and treating the previous and next chunk as a single continuum, making it easier to assess the actual impact of this architecture change.
3. A new dataset for benchmarking action-driven video generation models is introduced, leveraging existing datasets and adding new (mainly video game engine) data and labels. The authors compare their model with several s.o.t.a. models and introduce a new metric for scene consistency.
4. The model is benchmarked using several metrics, including one that requires multi-sampling with human labeling.

**Weaknesses:**

1. The interpretation of  fast and slow memory or training loops is a staple e.g. in generative replay in the continual learning domain (including often citing the hippocampus inspiration), as well as several RNN-like architectures with slow and fast memory. A link to these previous works (even if just in the appendix) could further improve the paper.
2. Novelty: the authors cite some previous work on the LLM domain as the main source of inspiration for this usage of temp-lora. The term temp-lora and its application in the video generation domain was used in previous works such as [1] (which is only a preprint, but given it is from feb 2024 still relevant for this review), a statement how their lora application differs would be appreciated. Given these works are only preprints it does not mean they take away from the novelty of this submission. It is just to get a better understanding of the differences in how lora is applied. Even if it is the same technique it would be interesting to know if the authors think the chosen task and datasets are better suited for the technique compared to the taks & datasets in the previous works.
3. I would like to see more architectural and (hyper) parameter details of the model used in the experiments (esp. The Unet + Lora), even if it is based on existing pretrained models, and how the hyperparmeters were chosen (if different from the base models).

Other inputs (not weaknesses) that may be used to improve the paper:
There seems to be repeating text in appendix A.2 Preliminaries On Low-Rank Adaptation

[1] Ren, Yixuan, et al. "Customize-a-video: One-shot motion customization of text-to-video diffusion models." arXiv preprint arXiv:2402.14780 (2024).

**Questions:**

1. Is there a limit of the temp-lora in terms of the capacity of trajectories of individual objects in the scene if their motion is not the same as the global motion?
2. I did not find the number of diffusion steps T used in the experiments, please provide these numbers and any other hyperparameters used if different than the base model.
3. How is the performance of the model on dynamic objects in a scene if you re-visit an area, e.g. moving objects like cars or persons in an early episode that you re-visit later?

---

> ### Author Response · Authors · 2024-11-19
> **Response to Reviewer iXgc (1/3)**
>
> *Thank you for your thoughtful review and strong endorsement of our work! We are delighted to address your questions by providing additional insights and carrying out additional experiments.*
>
> &nbsp;
>
> > **W1: The interpretation of fast and slow memory or training loops is a staple e.g. in generative replay in the continual learning domain (including often citing the hippocampus inspiration), as well as several RNN-like architectures with slow and fast memory. A link to these previous works (even if just in the appendix) could further improve the paper.**
>
> We sincerely thank the reviewer for this insightful suggestion about connecting to relevant prior work, and apologize for not adequately covering these important connections in our initial submission. We fully agree that works in continual learning and RNNs have provided fundamental inspirations for implementing slow-fast memory mechanisms. We have added an entire paragraph in the related works to discuss these papers in our revised paper draft. Please let us know if there are any further related works that we should cite in our paper. We welcome any suggestions for additional relevant works that would further enrich our discussion. We appreciate how this connection helps better position our work within the broader context of memory-based learning systems.
>
> &nbsp;
>
> > **W2: Novelty: the authors cite some previous work on the LLM domain as the main source of inspiration for this usage of temp-lora. The term temp-lora and its application in the video generation domain was used in previous works such as [1] (which is only a preprint, but given it is from feb 2024 still relevant for this review), a statement how their lora application differs would be appreciated. Given these works are only preprints it does not mean they take away from the novelty of this submission. It is just to get a better understanding of the differences in how lora is applied. Even if it is the same technique it would be interesting to know if the authors think the chosen task and datasets are better suited for the technique compared to the taks & datasets in the previous works.**
>
> We thank the reviewer for bringing up this interesting paper! We have cited this paper in our related works section.
>
> We would like to clarify that while sharing a similar name, the T-LoRA in [1] and our approach are **fundamentally different**:
> * **In terms of terminology**, the "temporal" in [1] refers to the temporal layers, and temporal-lora in [1] specifically refers to only tuning the temporal layers for modelling dynamics. On the contrary, the "temporal" in our context focuses on maintaining temporal consistency across long video sequences by dynamically updating LoRA parameters during inference to store and utilize episodic memory across distant chunks. Previously we thought about using "Memory-LoRA" to better illustrate our purpose, but later decided to stick to "Temp-LoRA" to pay tribute to the original Temp-LoRA mechanism introduced for long-context text generation.
> * **In terms of technical goals**, [1] focuses on motion customization in text-to-video models by using LoRA specifically on temporal attention layers to learn motion patterns from reference videos, while our TEMP-LORA is designed for storing episodic memory during inference time.
> * **In terms of architectures and mechanisms**, [1] applies LoRA to temporal attention layers in transformer architectures to capture motion signatures. It works only on a single chunk. Our approach applies LoRA to update the UNet in diffusion models, maintaining memory storage for a full episode consisting of multiple chunks.
> * **In terms of learning paradigms**, [1] uses a staged training pipeline for motion customization, while our approach implements a dual-speed learning system with memory consolidation.
>
> While these approaches differ substantially in their objectives and implementation, we find [1]'s innovative ideas highly inspiring for future extensions of our work. For instance, their motion adapter could potentially be integrated into our framework to better model dynamic objects and other agents in the scene, as the reviewer thoughtfully suggests in other questions. We appreciate this valuable connection!
>
> [1] Ren, Yixuan, et al. "Customize-a-video: One-shot motion customization of text-to-video diffusion models." arXiv preprint arXiv:2402.14780 (2024).

---

> ### Author Response · Authors · 2024-11-19
> **Response to Reviewer iXgc (2/3)**
>
> > **W3: I would like to see more architectural and (hyper) parameter details of the model used in the experiments (esp. The Unet + Lora), even if it is based on existing pretrained models, and how the hyperparmeters were chosen (if different from the base models).**
>
> Thank you for bringing this to our attention. We sincerely apologize for not providing sufficient information in the initial version.
>
> * **Architectural details**
> In the supplementary material, we initially included a detailed introduction to Modelscope. In the revised version, we have added Section A.3.2 in the appendix to provide additional architectural details. We hope this update enhances the understanding of our paper!
>
> * **Parameter details**
> We apologize that we did not include sufficient parameter details in the initial submission. We add additional details in the revised Supplementary Material Section C.1 Line 1099. Sorry about it!
> * **Parameter Choice**
> Sure! We are glad to share our attempts in choosing hyperparameters. Specifically, we carried out ablative studies on learning rate, lora rank and diffusion steps. We attach the results here as well as in the revised Appendix Section D.5.
>
> &nbsp;
>
> **Table A. Ablation Studies on Learning Rate and Lora Rank.**
> | Learning Rate | LoRA Rank | Scuts | SRC |
> |--------------|-----------|--------|-----|
> | lr=1e-3 | lora_rank=16 | 0.96 | 93.13 |
> | | lora_rank=32 | 1.35 | 92.87 |
> | | lora_rank=64 | 1.21 | 93.24 |
> | | lora_rank=128 | 1.40 | 92.53 |
> | lr=5e-4 | lora_rank=16 | 0.62 | 92.58 |
> | | lora_rank=32 | 0.48 | 93.20 |
> | | lora_rank=64 | 0.55 | 93.07 |
> | | lora_rank=128 | 0.59 | 92.87 |
> | lr=1e-4 | lora_rank=16 | 0.55 | 91.80 |
> | | lora_rank=32 | 0.37 | 93.71 |
> | | lora_rank=64 | 0.48 | 92.68 |
> | | lora_rank=128 | 0.52 | 92.45 |
> | lr=5e-5 | lora_rank=16 | 0.85 | 91.57 |
> | | lora_rank=32 | 0.92 | 91.98 |
> | | lora_rank=64 | 0.99 | 91.77 |
> | | lora_rank=128 | 1.04 | 91.44 |
> | lr=1e-5 | lora_rank=16 | 1.63 | 92.99 |
> | | lora_rank=32 | 1.25 | 94.74 |
> | | lora_rank=64 | 1.59 | 95.18 |
> | | lora_rank=128 | 1.65 | 93.97 |
> | Wo Temp-LoRA | | 1.88 | 89.04 |
>
> &nbsp;
>
> Table A presents the results of the ablation study on learning rate and LoRA rank. Our experiments show that the optimal parameter combination is lr=1e-4 and lora_rank=32, which yields the best performance. Consequently, we select this configuration for our final model. In practice, we observe that increasing the learning rate significantly leads to instability, causing the model to generate nonsensical videos in some cases. However, when the learning rate is too small, the model struggles to update the memory effectively, resulting in limited performance improvement compared to the model without Temp-LoRA.
>
> We also show the ablation studies on diffusion steps below, as well as in D.6, to demonstrate why we use 20 as the inference step.
>
> &nbsp;
>
> **Table B. Ablation Studies on inference-time diffusion steps**
> | Steps | FVD $\downarrow$ | PSNR $\uparrow$ | SSIM $\uparrow$ | LPIPS $\downarrow$ | SCuts $\downarrow$ | SRC $\uparrow$  |
> |-------|------------------|------------------|-----------------|-------------------|-------------------|----------------|
> | 10    | 595             | 17.82           | 57.21          | 27.89            | 0.45             | 90.12         |
> | 15    | 548             | **19.64**           | 59.15          | 26.32            | 0.41             | 91.95         |
> | 20| **514**         | 19.21       | **60.53**      | 25.06            | **0.37**         | 93.71     |
> | 25    | 519             | 19.08           | 60.12          | 24.98            | 0.38             | **93.85**         |
> | 30    | 527             | 18.95           | 59.87          | **24.92**            | 0.39             | 93.18         |
> | 50    | 542             | 18.73           | 59.24          | 25.15            | 0.42             | 92.54         |
> | 75    | 568             | 18.41           | 58.65          | 25.45            | 0.44             | 91.87         |
> | 100   | 589             | 18.12           | 58.12          | 25.82            | 0.46             | 91.23         |

---

> ### Author Response · Authors · 2024-11-20
> **Response to Reviewer iXgc (3/3)**
>
> > **Q1 & Q3: Is there a limit of the temp-lora in terms of the capacity of trajectories of individual objects in the scene if their motion is not the same as the global motion? How is the performance of the model on dynamic objects in a scene if you re-visit an area, e.g. moving objects like cars or persons in an early episode that you re-visit later?**
>
> Thank you for the insightful question!
> * **Current Data Coverage**
>     Our current dataset (200k videos) primarily focuses on single-agent scenarios, which is appropriate for the current stage of research. We acknowledge this limitation and plan to expand our dataset to include more multi-agent and dynamic object scenarios. However, we believe this data expansion is orthogonal to our core contribution, and can be seamlessly integrated without requiring significant changes to the model architecture or learning process.
> * **Additional Experiments**
>     As suggested by Reviewer C2ZJ, we carry out additional experiments on a real-world dataset *Walking Tour*, which captures a large number of objects and actions from first-person view. We attach the qualitative results in [the rebuttal visualization website](https://sites.google.com/view/slowfastvgen/home) Section 5.
>
>
> From the results, we can see that because currently we do not explicitly model the actions of other agents, they tend to follow random trajectories that conform with their intial pose.
> * **Scene Revisiting with dynamic objects**
>     We further carry out scene revisiting experiments using Minecraft environments where there are animals and skeletons, and show quantitative results in [the rebuttal visualization website](https://sites.google.com/view/slowfastvgen/home) Section 6. We observe that the other agents tend to take random actions (example 1). However, when the dynamic objects become too small or hidden by trees, they may gradually disappear (example 2). We observe that the movements of dynamic objects in Minecraft are often unpredictable and challenging to model—for instance, an animal suddenly appearing from behind a tree. This unpredictability makes the problem both complex and intriguing, highlighting its potential for future exploration.
>
> * **Thoughts about Future Integration of Multi-Agent Settings**
>     We envision two promising approaches to handle dynamic objects:
>     * Default Motion Modeling: Background objects could follow learnable default behaviors (e.g., cars following traffic patterns, pedestrians walking on sidewalks). This could be achieved by learning motion priors from our diverse dataset and applying them to maintain scene consistency during revisits.
>     * Multi-agent Action Control: We could extend our framework to explicitly model multiple agents, where each agent has its own TEMP-LORA module for memory storage. This would allow independent control and memory maintenance for each dynamic object while preserving global scene consistency.
>
> Our slow-fast learning approach naturally facilitates these extensions through its episodic memory mechanism. The long-term memory storage enables early action recognition and proactive planning based on motion priors (*e.g.*, anticipating a thrown object's trajectory or predicting pedestrian movements). Moreover, since our model has demonstrated strong performance in single-agent scenarios, multi-agent control could be elegantly achieved by composing multiple TEMP-LORA modules, each maintaining memory for a specific agent.
>
> The integration of motion modeling techniques from [1] could further improve dynamic object representations, while existing multi-agent world models could enhance coordination capabilities. We believe this direction holds great promise for handling complex, dynamic real-world scenarios.
>
> &nbsp;
>
> > **Q2: I did not find the number of diffusion steps T used in the experiments, please provide these numbers and any other hyperparameters used if different than the base model.**
>
> We train with 1000 diffusion steps following DDPM and perform inference using DPM-Solver  with 20 steps and classifier-free guidance. We apologize that we did not make this clear in the initial draft! We include this in the revised Appendix line 1100, with additional parameters such as the guidance scale.
>
> &nbsp;
>
> *We once again thank your efforts in reviewing our paper! Your valuable comments really help improve our paper, and inspire us a lot! We hope our response has addressed your concerns. If you have more questions, don't hesitate to let us know in the rebuttal window! Thanks again!*
>
> &nbsp;
>
> Best,
> Authors

---

> ### Comment · Reviewer_iXgc · 2024-11-25
>
> Thank you to the authors for this detailed response and for considering the reviewer's input! The provided additional details such as the experimental parameters and comparison to other research helped to better understand the approach.

---

### Official Review · Reviewer_C2ZJ · 2024-11-08

**Soundness:** 3
**Presentation:** 4
**Contribution:** 3
**Rating:** 8
**Confidence:** 4

**Summary:**

In this paper, the authors focus on the problem of long term video generation. To this end, they propose SlowFast-VGen, a model which leverages episodic memory (using a LoRA module from long text generation) to improve context utilization over longer time horizons. The authors also argue that the approach has connections to Cognitive Science. They then evaluate their approach on multiple datasets including RLBench and Minecraft. They demonstrate convincing quantitative results that beat baselines on metrics measuring long term consistency.

**Strengths:**

1. The paper addresses a serious failing in contemporary approaches to video modeling. Current video models cannot scale well to long time horizons, and this paper shows promise in advancing this capability.

2. The paper is well written and easy to understand.

3. Experimental evaluation is thorough, and the datasets used is reasonably diverse. The quantitative results are convincing.

**Weaknesses:**

1. The proposed method is quite complex, requiring multiple working parts (Slow vs Fast modules).

2. I think the paper might benefit from more complex datasets. Datasets from video games, robotic simulation, and car driving are used, but these are not particularly diverse. Training/testing on a dataset such as Walking Tours (Venkataramanan et al 2024) would strengthen the paper.

**Questions:**

1. The approach is complex. Are there any possible ways to streamline the model while still leveraging the Slow/Fast concept?

2. How would this approach deal with a complex "real world" dataset of long videos beyond just constrained domains such as Kitchens or Car Driving?

---

> ### Author Response · Authors · 2024-11-19
> **Response to Reviewer C2ZJ**
>
> *Thank you for your thorough review and the positive rating of our work! We greatly appreciate your insightful and constructive questions that have helped us reflect on and clarify important aspects of our approach.*
>
> &nbsp;
>
> > **W1 & Q1 The approach is complex. Are there any possible ways to streamline the model while still leveraging the Slow/Fast concept?**
>
> We appreciate the reviewer's concern. We acknowledge that our paper's presentation of the approach could be simplified and apologize for any complexity in the narration. We will improve the clarity and simplify the explanation in the final version.
>
> However, we would like to also emphasize that the core idea and design of our approach are actually simple, clean, effective, and elegant:
> * Our approach builds on a standard yet efficient conditional video diffusion model with two sets of parameters: base parameters $\Phi$ for slow learning across diverse scenarios, and LoRA parameters $\Theta$ for fast learning in specific episodes
> * The two learning phases operate in perfect symmetry: slow learning freezes $\Theta$ to update $\Phi$, while fast learning freezes $\Phi$ to update $\Theta$, forming a natural and elegant iterative loop
> * The implementation is remarkably lightweight - TEMP-LORA for fast learning only introduces a small set of parameters during inference (increasing memory by just 352MB)
> * This minimal yet powerful design naturally mirrors the complementary learning systems in human cognition, where slow neocortical learning harmoniously interacts with fast hippocampal memory formation
>
> We apologize again that we did not clarify this method more concisely in the paper. We will improve our presentation while maintaining the key insights in our final version. Thank you for pointing this out!
>
> &nbsp;
>
> > **W2 & Q2 I think the paper might benefit from more complex datasets. Datasets from video games, robotic simulation, and car driving are used, but these are not particularly diverse. Training/testing on a dataset such as Walking Tours (Venkataramanan et al 2024) would strengthen the paper. How would this approach deal with a complex "real world" dataset of long videos beyond just constrained domains such as Kitchens or Car Driving?**
>
> * **Integration of Walking Tours** Thank you so much for this suggestion! *Walking Tours* is such a wonderful paper, and we have already cited it in our revised version! We also run **additional experiments on walking tour**. We attach the experimental results below in Table A as well as in the revised Apendix Section D.4. We also includes qualitative examples in [the rebuttal visualization website](https://sites.google.com/view/slowfastvgen/home) Section 5.
>
> &nbsp;
>
> **Table A. Results on Walking Tour**
> Model | FVD ↓ | PSNR ↑ | SSIM ↑ | LPIPS ↓
> ---|---|---|---|---
> AVDC | 1678 | 15.12 | 48.90 | 27.45
> Streaming-T2V | 1234 | 13.89 | 45.67 | 35.78
> Runway Gen-3 Turbo | 1989 | 10.78 | 44.56 | 54.90
> AnimateDiff | 998 | 16.45 | 49.89 | 36.78
> SEINE | 1156 | 16.78 | 51.23 | 37.90
> iVideoGPT | 1567 | 12.45 | 29.90 | 29.89
> VideoGPT+ | 1445 | 13.90 | 43.67 | 31.23
> Ours SlowFast-VGen | 767 | 17.89 | 56.78 | 23.90
>
> &nbsp;
>
> We can see that our model does have the potential to deal with complex real-world data. For future development, we see great potential in augmenting Walking Tours with complex action annotations. We can use motion detectors to detect the motions and segment the video, while having ChatGPT come up with language action annotations of the given duration. We believe this dataset holds immense promise, as it captures both authentic real-world scenes and natural movements. Thank you once again for introducing such a valuable resource!
>
> * **Clarification of Diversity** We acknowledge the limitations in scene coverage, as explicitly stated in Section 5 of our paper, which is why we carefully position ourselves as an "approximate world model". However, we would like to respectfully note that our dataset's diversity significantly exceeds many existing approaches that claim to be world models that were trained exclusively on robot manipulation data. Our training set encompasses not only robotics but also Unreal environments, real-world driving scenarios, and human activities, representing a substantial step toward more comprehensive world modeling.
>
> &nbsp;
>
> *We sincerely thank you for your positive assessment of our work! Your questions have helped us articulate the strengths of our approach more clearly, and we will incorporate these clarifications in the final version. If you have any additional questions, we would be happy to address them during the rebuttal window.*
>
> &nbsp;
>
> Best,
> Authors

---

### Official Review · Reviewer_iFRv · 2024-11-09

**Soundness:** 3
**Presentation:** 3
**Contribution:** 3
**Rating:** 6
**Confidence:** 3

**Summary:**

The paper introduces SLOWFAST-VGEN, a dual-speed learning framework for action-driven, long video generation inspired by the complementary learning system in human cognition. This model integrates slow learning of world dynamics with fast learning for episodic memory storage, aiming to address challenges in generating consistent, coherent long videos. The slow learning component utilizes a masked conditional video diffusion model, while the fast learning component leverages LoRA to store episodic memory. The authors evaluate the model on various datasets and metrics, achieving superior FVD scores and scene consistency compared to baseline models.

**Strengths:**

1. **Novelty in Dual-Speed Learning**: By combining slow learning of world models and fast episodic learning, SLOWFAST-VGEN represents an innovative approach, mimicking cognitive memory systems effectively.
2. **Comprehensive Benchmarking**: Evaluations include diverse tasks and metrics, reinforcing the model’s robustness across applications.

**Weaknesses:**

1. **Limited Generalization**: SLOWFAST-VGEN’s performance may drop in scenarios outside its training domains, such as novel environments or complex, unforeseen actions, limiting its adaptability in dynamic real-world applications.
2. **Memory Inconsistency Over Long Sequences**: TEMP-LORA effectively stores episodic memory within single episodes but struggles with consistency across multi-episode or long-term tasks, potentially leading to fragmented recall in extended sequences. Since the authors mention that the slowing learning for world modeling can capture general dynamics, some long-horizon tasks need to be evaluated, such as moving to the right and then to the left.
3. **High Computational Cost**: The reliance on a diffusion model for video generation demands significant computational resources. Although using LoRA, the two-stage training. What about latent diffusion models for the slow learning, while video diffusion models for the fast learning?

**Questions:**

1. How are the parameters optimized across the slow and fast learning phases to ensure smooth interaction without redundancy? Could alternative approaches, such as dynamically adjusting memory encoding based on the video’s complexity, further streamline the integration?
2. For tasks that require nuanced, multi-step interactions (e.g., tasks involving conditional actions or dependencies between actions), how does SLOWFAST-VGEN manage action dependencies across time steps? Are there specific mechanisms to ensure that complex sequences remain consistent and contextually appropriate?
3. What do you think about the hierarchical learning paradigm? Such as the slow/fast learning v.s. long-horizon key-frame generation + show-horizon dense-frame generation. This is just a question, not need to conduct experiments.

---

> ### Author Response · Authors · 2024-11-19
> **Response to Reviewer iFRv (1/4)**
>
> *We appreciate the positive and constructive comments from you, which are crucial for improving the paper! We have updated the paper based on your suggestions and carried out additional experiments to resolve your concerns.*
>
> &nbsp;
>
> > **W1: Limited Generalization. SLOWFAST-VGEN’s performance may drop in scenarios outside its training domains, such as novel environments or complex, unforeseen actions, limiting its adaptability in dynamic real-world applications.**
>
>
> We sincerely appreciate the reviewer raising this important concern about generalization. We would like to address this point by highlighting several aspects of our work:
>
> * **Strong Generalization to Unseen Scenes**
>     * We want to emphasize that all our experimental evaluations are conducted exclusively on **unseen test scenes that have no overlap with the training data**, providing a rigorous test of generalization, which was also emphasized at Line 412 and Line 468 in our paper.
>     * The fast learning mechanism, inspired by the human hippocampus's role in rapid learning, is specifically designed for handling quick adaptation to new environments during inference time. It enables real-time parameter adaptation to encode and utilize episodic memories of the current scene during inference.
>     * This can be observed in Figure 4, 9 and 10, where our model successfully handles previously unseen environments while maintaining consistency across an extended sequence, demonstrating its adaptability to novel settings. The quantitative results also support this, with our model achieving superior performance on unseen scenarios (FVD score of 514 compared to baselines >780).
>     * We apologize that we did not emphasize this generalization ability enough in the initial draft. We have added extra highlights in Line 478 of the revised version about this. Thanks again for pointing this out!
>
> * **Action Generalization Capabilities**
>     * Although generalization over unseen actions was not specifically tackled in our initial draft, we provide insights into how action generalization could be achieved under the current framework. Specifically, by leveraging CLIP's open-vocabulary capabilities for action encoding, our model could handle novel actions through: 1) Interpolation in the CLIP embedding space between semantically related actions (e.g., between "walk" and "run" to generate "jog", or between "lift" and "raise" for novel manipulation descriptions) 2) Transfer of similar actions across different contexts thanks to CLIP's semantic understanding (*e.g.*, "go right" of humans and cars could be generalized to "Steer the bicycle to the right".) 3) Composition of basic actions into more complex sequences (*e.g.*, combining "pick up", "move", "put down" with unseen objects to complete novel manipulation tasks).
>     * To further demonstrate this, we conduct **additional experiments on action generalization** on these three perspectives, and show the results below (Table A) as well as in revised Appexdix Section D.2 (in blue). We also show **qualitative results** in [the anonymous rebuttal website](https://sites.google.com/view/slowfastvgen/home)  Section 4.
>
> &nbsp;
>
> **Table A. Action Generalization Results**
> |                      | FVD | PSNR  | SSIM  | LPIPS |
> |----------------------|-----|-------|-------|-------|
> | Action Interpolation | 692 | 17.25 | 55.26 | 26.90 |
> | Similar Actions      | 579 | 18.18 | 57.72 | 28.74 |
> | Action Composition   | 816 | 16.97 | 51.35 | 34.54 |
> | Original             | 514 | 19.21 | 60.53 | 25.06 |
>
> &nbsp;
>
> &emsp;&emsp; &emsp; From the results, we can see that our model has action generalization abilities to a great extent, with similar action transfer performing best (FVD=579) due to CLIP's strong semantic understanding, while action interpolation shows moderate degradation (FVD=692). Action composition proves most challenging (FVD=816), suggesting that combining multiple basic actions introduces complexity that the current framework struggles to handle seamlessly.
>
>
> * While we acknowledge that performance may vary in extremely novel scenarios (as with any machine learning system), our diverse training data, fast learning mechanism, and continuous action embedding space through CLIP provide a strong foundation for generalization.
>
> We appreciate the reviewer's concern and will continue to improve generalization in future work.

---

> ### Author Response · Authors · 2024-11-19
> **Response to Reviewer iFRv (2/4)**
>
> > **W2: Memory Inconsistency Over Long Sequences. TEMP-LORA effectively stores episodic memory within single episodes but struggles with consistency across multi-episode or long-term tasks, potentially leading to fragmented recall in extended sequences. Since the authors mention that the slowing learning for world modeling can capture general dynamics, some long-horizon tasks need to be evaluated, such as moving to the right and then to the left.**
>
>
> Thank you for raising this concern and giving us a chance to clarify our novelties.
> We want to emphasize that our model naturally demonstrates strong memory consistency across both single long episodes and multi-episode sequences through two key mechanisms:
> * **Long-term Memory Storage by Design.**
>     Our fast learning mechanism is specifically tailored to handle **long-term episodes  by storing episodic memory of the entire trajectory in TEMP-LORA parameters, not just recent chunks**. We have evaluated numerous long-horizon tasks, with many qualitative examples specifically demonstrating consistent bidirectional navigation (moving left/right, forward/back) across different environments, as shown in Figures 4, 5, 9, 10. These examples clearly demonstrate that our model maintains consistency across the full sequence, compared to both baselines and our model without fast learning mechanism. Our model can maintain consistency in extended sequences of up to **1000 frames**, which is quantitatively validated through Scene Cuts (SCuts) of 0.37 compared to baselines >0.89 and Scene Return Consistency (SRC) of 93.71% compared to baselines <91%. In fact, contrary to concerns raised, **maintaining memory consistency over long sequences is one of the key strengths of our approach**.
> * **Multi-episode Learning and Memory Consolidation.**
>     Our slow-fast learning loop algorithm (Section 3.3) specifically addresses consistency across multiple long-term episodes through memory consolidation from the fast learning phase into the slow learning process. We validate this through carefully designed tasks (Section 4.2) that specifically test skill learning through multi-episode memory: Robot Manipulation (moving objects and returning them to original positions) and Game Navigation (retracing complex paths to return to starting points). Both tasks demonstrate superior performance as shown in Tables 2 and Figures 5. These results directly validate our model's ability to maintain consistency in long sequence generation with multi-directional navigation, precisely addressing the reviewer's concern.
>
> We apologize that we did not make these contributions sufficiently clear in our original submission. We promise that we will improve the manuscript on this perspective.
>
> &nbsp;
>
> > **W3: High Computational Cost: The reliance on a diffusion model for video generation demands significant computational resources. Although using LoRA, the two-stage training. What about latent diffusion models for the slow learning, while video diffusion models for the fast learning?**
>
> We sincerely appreciate that you have pointed this out, and we apologize for any misunderstanding that may have been caused.
>
> * We would like to clarify that our model **already uses latent diffusion for both slow and fast learning**, operating in the compressed latent space of a pre-trained VAE as detailed in Section 3.1, 3.2 and Appendix A.1. We respectfully note that using different architectures for slow and fast learning as suggested would actually increase model complexity and memory usage, as we would need to maintain two separate diffusion models. We appreciate the reviewer bringing this up and will better emphasize these design choices in the final version of the paper.
>
> * The fast learning phase is lightweight by design, as the TEMP-LORA module introduces minimal computational overhead during inference (only 9931MB vs. 9579MB memory usage, and 13.81s vs. 12.93s per sample, as shown in Table 3 of Supplementary Materials)

---

> ### Author Response · Authors · 2024-11-19
> **Response to Reviewer iFRv (3/4)**
>
> > **Q1: How are the parameters optimized across the slow and fast learning phases to ensure smooth interaction without redundancy? Could alternative approaches, such as dynamically adjusting memory encoding based on the video’s complexity, further streamline the integration?**
>
> We thank the reviewer for this deep insight!
>
> * **Parameter Optimization Across Phases**
> The two phases are orthogonal by design, as we freeze the base model parameters $\Phi$ when updating TEMP-LORA parameters $\Theta$ during fast learning, and freeze TEMP-LORA parameters when updating base parameters during slow learning, as shown in Algorithm 1(b). They are also trained on different objectives with distinct purposes: slow learning focuses on capturing general world dynamics across diverse scenarios through pre-training, while fast learning specifically tackles episodic memory storage for the current scene during inference. This orthogonal design ensures clear separation of concerns and prevents redundancy between the two phases.
>
> * **Dynamic Memory Encoding Based On Video Complexity**
> Thank you for this insightful suggestion! Inspired by this, we have conducted comprehensive ablation studies exploring the impact of LoRA rank and the fast learning rate across three test sets with varying video complexity (simple, medium, and complex).  We report the performance below as well as in Appendix Section D.3 (in blue).
>
> &nbsp;
>
> **Table B. Parameter Selection based on video complexity**
>
> | Learning Rate | LoRA Rank | Single Set |  | Original Set |  | Complex Set |  |
> |--------------|-----------|-------------|------------|--------------|------------|-------------|------------|
> |              |           | Scuts | SRC | Scuts | SRC | Scuts | SRC |
> | lr=1e-3      | 16        | 0.80  | 89.91 | 0.96 | 93.13 | 0.69 | 93.56 |
> |              | 32        | 1.99  | 85.83 | 1.35 | 92.87 | 0.65 | 95.44 |
> |              | 64        | 3.52  | 60.72 | 1.21 | 93.24 | 0.64 | 94.47 |
> | lr=1e-4      | 16        | 0.15  | 94.78 | 0.55 | 91.80 | 0.75 | 91.10 |
> |              | 32        | 0.15  | 95.56 | 0.37 | 93.71 | 0.72 | 91.15 |
> |              | 64        | 0.24  | 93.33 | 0.48 | 92.68 | 0.70 | 92.35 |
> | lr=1e-5      | 16        | 0.17  | 95.51 | 1.63 | 92.99 | 1.91 | 90.01 |
> |              | 32        | 0.14  | 95.54 | 1.25 | 94.74 | 1.84 | 89.38 |
> |              | 64        | 0.15  | 93.69 | 1.59 | 95.18 | 1.57 | 89.92 |
> | Wo Temp-LoRA |           | 0.81  | 95.13 | 1.88 | 89.04 | 1.89 | 89.49 |
>
>
> &nbsp;
>
>
> Based on the ablation results, we observe that the optimal hyperparameters vary across different scene complexities, suggesting the potential benefit of dynamic adjustment strategies. For simpler scenes, a smaller learning rate (1e-5) with moderate LoRA rank (32) achieves the best scene consistency (SCuts: 0.14), as these scenes require more careful memory updates. For our original test set with medium complexity, a moderate learning rate (1e-4) with rank 32 provides the best balance (SCuts: 0.37, SRC: 93.71). Interestingly, more complex scenes benefit from a larger learning rate (1e-3) and higher LoRA rank (64) to capture more intricate details (SCuts: 0.64, SRC: 94.44).
>
> We also observe that performance collapses under certain conditions - notably with high learning rate (1e-3) and high rank (64) in simple scenes (SCuts: 3.52, SRC: 60.72). This is likely due to overfitting to specific scene details rather than capturing general memory patterns. For complex scenes, small learning rates (1e-5) result in poor performance (SCuts: >1.5) comparable to no training at all, as the model fails to adequately capture the complex scene dynamics. These findings suggest that adapting hyperparameters based on scene complexity could further improve performance, and we plan to explore automatic adjustment mechanisms in future work. Actually, we find our parameter set (lr = 1e-4, lora_rank = 32) provides the most stable performances across different scenarios, and that's why we choose them.
>
> While dynamic memory allocation could benefit variations on scene complexities, the challenge can lie in how to assess the complexity of a scene (*e.g.,* using scene graph size or pixel differences?). Also, how to create a mapping from the scene complexity and the detailed parameter combinations is worth delving into in the future.
>
> We once again thank you for this profound insight!

---

> ### Author Response · Authors · 2024-11-19
> **Response to Reviewer iFRv (4/4)**
>
> > **Q2:For tasks that require nuanced, multi-step interactions (e.g., tasks involving conditional actions or dependencies between actions), how does SLOWFAST-VGEN manage action dependencies across time steps? Are there specific mechanisms to ensure that complex sequences remain consistent and contextually appropriate?**
>
> We appreciate the reviewer's question about handling multi-step interactions. Our framework addresses complex dependencies through two complementary mechanisms:
> * **Memory-augmented Dependency Learning**
> The slow-fast learning loop naturally handles action dependencies through its dual-phase structure. In the fast learning phase, TEMP-LORA maintains long-context dependencies by storing the full trajectory history. Slow learning thus has access to both TEMP-LORA parameters and ground-truth videos, allowing it to learn valid action sequences and their dependencies.
> * **Physics-aware Action Generation**
> During slow learning on massive data, the model learns general physics rules as well as recognizing unexecutable actions and their consequences. e.g., In Figure 3, when the agent encounters physical barriers like fences, it naturally learns to stop rather than attempting impossible actions. If a prerequisite action fails, the model appropriately stops execution of dependent actions. This built-in safety mechanism, learned from real-world physical constraints in our training data, helps prevent invalid action sequence
>
> &nbsp;
>
> > **Q3:What do you think about the hierarchical learning paradigm? Such as the slow/fast learning v.s. long-horizon key-frame generation + show-horizon dense-frame generation. This is just a question, not need to conduct experiments.**
>
> Good question!
>
> We believe that while key-frame + dense-frame generation approach is valuable for certain video synthesis tasks, it faces several key challenges. We also covered these challenges in related works in the initial submission (Line 149 to Line 154)
>
> * The requirement of predefined key-frames makes it unsuitable for stream-in scenarios like gaming applications where actions are received and executed in real-time
> * The approach deviates from how human beings process and store sequential memory
> * The method is constrained by the limited availability of long video sequences with key frames and action annotations for training, while our approach can effectively learn from shorter videos through slow learning.
>
> In contrast, our slow-fast learning approach offers several advantages:
>
> * More suitable for real-time, interactive scenarios where actions are streamed in
> * Better maintains action-state correspondence through direct conditioning
> * Naturally handles variable-length sequences without predefined key-frames
> * Stores memory of a long trajectory in efficient LoRA parameters, therefore could be used for skill learning in slow-fast learning loop.
>
> We appreciate the reviewer for bringing this up and are glad to have the opportunity to further highlight the advantages of our model over hierarchical learning models.
>
> &nbsp;
>
> *Thank you again for your thoughtful questions that have helped us clarify key aspects of our work! We hope our detailed responses have addressed your concerns and further demonstrated the strengths of our approach, and therefore have reinforced your confidence in our work. We will incorporate these clarifications in the final version to better present our contributions.  If you have any additional questions, we would be happy to address them during the rebuttal window. Thanks again!*
>
> &nbsp;
>
> Best,
> Authors

---

> > ### Author Response · Authors · 2024-12-02
> > **Looking forward to your post-rebuttal reply!**
> >
> > Dear Reviewer,
> >
> > Thank you once again for your thoughtful feedback on our paper. As the discussion period draws to a close, we want to ensure that our responses have adequately addressed your concerns. If so, would you kindly consider raising the score? Please also let us know if there are any remaining questions or points that require further clarification, or revision in the camera-ready version.
> >
> > We appreciate your time and effort in reviewing our work.
> >
> > Best regards,
> > The Authors

---

> ### Author Response · Authors · 2024-11-25
> **Follow-up on rebuttal**
>
> Dear Reviewer,
>
> Thanks again for your suggestions to strengthen this work! As the rebuttal period is approaching the end soon, we want to know if our response has answered your questions and addressed your concerns. If no, we are more than happy to provide further modifications. If yes, would you kindly consider raising the score?
>
> Thanks again for your truly constructive and insightful feedback.
>
> Best, Authors

---

### Author Response · Authors · 2024-11-20
**General Response to All Reviewers**

We sincerely appreciate all reviewers’ time and efforts in reviewing our paper. In addition to the response to specific reviewers, here we would like to highlight our contributions and the new experiments that we add in the rebuttal.

**[Our Contributions]**
We are glad to find out that the reviewers generally acknowledge our contributions:
* The paper is well-written and easy to follow [C2ZJ, iXgc, bXBH]
* The paper introduces a novel bio-inspired interpretation of a slow-and fast learning loop to improve long video generation by maintining an episodic memory [iFRv, C2ZJ, iXgc, bXBH]
* The authors introduce a collection of action-conditioned video that could be useful for future related research [iFRv, C2ZJ, iXgc, bXBH]
* Experimental evaluation is thorough [iFRv, C2ZJ, iXgc, bXBH]

**[New Experiments]**
In this rebuttal, we have added more supporting experiments to address reviewers’ concerns.
* Results on Action Generalization (iFRv)
* Parameter Selection based on video complexity (iFRv)
* Results on the Walking Tour dataset (C2ZJ)
* Ablation Studies on Learning Rate and Lora Rank (iXgc)
* Ablation Studies on Inference-Time Diffusion Steps (iXgc)
* Experimental Results on the Teco-Habitat Dataset (bXBH)

**[Qualitative examples]**
We include our qualitative examples of all given experiments in https://sites.google.com/view/slowfastvgen/home (Please use a computer to open the demo website. If prompted "sign up needed", you may close it and reopen after a while, or sign in to your google account, or click copy link and paste in a new browser window.)
* Examples on Action Generalization (Section 4)
* Examples on Walking Tour (Section 5)
* Examples on Scene Revisit with Dynamic Objects (Section 6)
* Action-Conditioned Long Video Generation on Teco-Habitat (Section 7)

We hope our responses below convincingly address all reviewers’ concerns. We thank all reviewers’ time and efforts again!

---

### Author Response · Authors · 2024-11-25
**Thank you and we are looking forward to your post-rebuttal feedback!**

Dear AC and all reviewers:

Thanks again for all the insightful comments and advice, which helped us improve the paper's quality and clarity.

We would love to convince you of the merits of the paper. Please do not hesitate to let us know if there are any additional experiments or clarification that we can offer to make the paper better. We appreciate your comments and advice.

Best,

Author

---

### Meta-Review · Area_Chair_wPq7 · 2024-12-18

**Metareview:**

This paper introduces SlowFast-VGen, a novel dual-speed learning system for action-driven long video generation inspired by human cognitive learning systems. The paper received strong scores (6,8,8,8) and positive recommendations from all reviewers, with consistent praise for its innovative approach to addressing long-term consistency in video generation.
The reviewers highlighted several key strengths: the bio-inspired slow-fast learning framework that effectively maintains episodic memory, strong empirical results showing significant improvements over baselines (FVD score of 514 vs 782), and the creation of a valuable large-scale action-annotated video dataset. The paper was also commended for its clear presentation and thorough experimental evaluation.

Initial concerns focused on three main areas: the generalization capabilities to novel scenarios, the complexity of the approach, and questions about handling more challenging long-term dependencies. The authors provided comprehensive responses, including additional experiments on the Walking Tour dataset showing strong generalization, clarification of the model's elegant core design despite its apparent complexity, and new results on the Teco-Habitat dataset demonstrating promising capabilities for complex navigation tasks.

While some reservations remain about the model's ability to learn abstract long-term representations (as noted by reviewer bXBH), there is consensus that the paper represents an important step forward in test-time training for long video generation. The authors' rebuttal thoroughly addressed most concerns, leading one reviewer to explicitly raise their score.

**Additional Comments On Reviewer Discussion:**

None -- see metareview

---

### Decision · Program_Chairs · 2025-01-22

Accept (Spotlight)